

**Assessment of Simulated Soil Moisture from WRF Noah, Noah-MP,**
**and CLM Land Surface Schemes for Landslide Hazard Application**
Lu Zhuo[1], Qiang Dai[1,2*], Dawei Han[1], Ningsheng Chen[3], Binru Zhao[1,4]
[1]WEMRC, Department of Civil Engineering, University of Bristol, Bristol, UK
[2]Key Laboratory of VGE of Ministry of Education, Nanjing Normal University, Nanjing, China
[3]The Institute of Mountain Hazards and Environment (IMHE), China
[4]College of Water Conservancy and Hydropower Engineering, Hohai University, Nanjing, China
*Correspondence: civengwater@gmail.com
**Abstract**
This study assesses the usability of Weather Research and Forecasting (WRF) model simulated
soil moisture for landslide monitoring in the Emilia Romagna region, northern Italy during the 10-
year period between 2006 and 2015. Particularly three advanced Land Surface Model (LSM)
schemes (i.e., Noah, Noah-MP and CLM4) integrated with the WRF are used to provide
comprehensive multi-layer soil moisture information. Through the temporal evaluation with the
in-situ soil moisture observations, Noah-MP is the only scheme that is able to simulate the large
soil drying phenomenon close to the observations during the dry season, and it also has the highest
correlation coefficient and the lowest *RMSE* at most soil layers. Each simulated soil moisture
product from the three LSM schemes is then used to build a landslide threshold model, and within
each model, 17 different exceedance probably levels from 1% to 50% are adopted to determine
the optimal threshold scenario (in total there are 612 scenarios). Slope degree information is also
used to separate the study region into different groups. The threshold evaluation performance is
based on the landslide forecasting accuracy using 45 selected rainfall events between 2014-2015.
Contingency tables, statistical indicators, and Receiver Operating Characteristic analysis for
different threshold scenarios are explored. The results have shown that the slope information is
very useful in identifying threshold differences, with the threshold becoming smaller for the

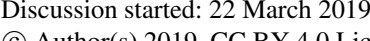
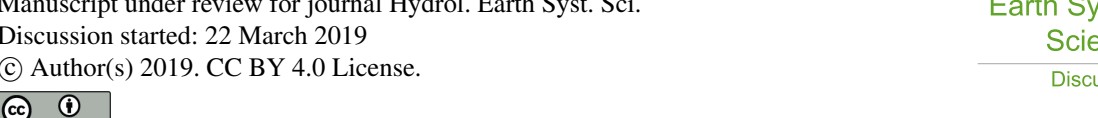


steeper area. For landslide monitoring, Noah-MP at the surface soil layer with 30% exceedance
probability provides the best landslide monitoring performance, with its hitting rate at 0.769, and
its false alarm rate at 0.289.
**Keywords:** Emilia Romagna, Weather Research and Forecasting (WRF) Model, Land Surface
Model (LSM), Numerical Weather Prediction (NWP) model, landslide hazards, soil moisture.
**1.  Introduction**
Landslide is a repeated geological hazard during rainfall seasons, which causes massive
destructions, loss of lives, and economic damages worldwide (Klose et al., 2014). It is estimated
between 2004 and 2016, there is a total number of 4862 fatal non-seismic landslides occurred
around the world, which had resulted in the death of over 55,000 people (Froude and Petley, 2018).
Those numbers are expected to further increase due to extreme events induced by climate changes
and pressures of population expanding towards unstable hillside areas (Gariano and Guzzetti,
2016;Petley, 2012). The accurate predicting and monitoring of the spatiotemporal occurrence of
the landslide is the key to prevent/ reduce casualties and damages to properties and infrastructures.
The most widely adopted method for real-time landslide monitoring is based on the simple
empirical rainfall threshold, which has been used in many countries for their national landslide
early warning system (Caine, 1980). The method commonly relies on building the rainfall
intensity-duration curve using the information from the past landslide events (Chae et al., 2017).
However, such a method in many cases is insufficient for landslide hazard assessment (Posner and
Georgakakos, 2015), because in addition to rainfall, initial soil moisture condition is one of the
main triggering factors of the events (Glade et al., 2000;Crozier, 1999;Tsai and Chen, 2010;Hawke
and McConchie, 2011;Bittelli et al., 2012;Segoni et al., 2018;Valenzuela et al., 2018;Bogaard and
Greco, 2018).





Although some researches have recognised the significance of soil moisture information for
landslide early warning, most of them only rely on the antecedent precipitation index which simply
depends on the amount of total rainfall accumulated before a landslide event occurs (Chleborad,
2003;Calvello et al., 2015;Zêzere et al., 2005). Such a method is not recommended by several
studies (Pelletier et al., 1997;Baum and Godt, 2010;Brocca et al., 2008), because during wet
seasons, soil is often already saturated, and any additional rainfall falls on the earth surface will
become direct runoff (Zhuo and Han, 2016b, a). As a result, the antecedent precipitation method
can sometimes significantly overestimate the soil wetness condition. On the other hand,
evapotranspiration is another factor which controls the soil moisture temporal evolution, which
can also influence the relationship between the actual and the estimated soil moisture. Therefore,
it is important that the landslide hazard assessment should be based on the real soil moisture
information.
Soil moisture varies largely both spatially and temporally (Zhuo et al., 2015b). For landslide
applications, to accurately monitor soil moisture fluctuations in a critical zone (normally in remote
regions), a dense network of soil moisture sensors is prerequisite. However, because of the
complex installation and high maintenance fee especially in steep mountainous areas, such
networks are normally unavailable. Several studies have found the usefulness of ground-measured
soil moisture data for landslide monitoring purpose (Greco et al., 2010;Baum and Godt,
2010;Harris et al., 2012;Hawke and McConchie, 2011). However, due to the sparse distribution/no
existence of in-situ sensors in most hazardous regions, alternative soil moisture data sources need
to be explored.  One of the data sources is through satellite remote sensing technologies. Although
such technologies have been improved significantly over the past decade (Zhuo et al., 2016a), their
retrieving accuracy is still largely affected by meteorological conditions (cloud coverage and



rainfall), frozen soil conditions (Zhuo et al., 2015a), and dense vegetation coverages particularly
in mountainous regions (Temimi et al., 2010); furthermore, the acquired data only covers the top
few centimetres of soil, and their resolution is too low (e.g., 0.25 degree) for detailed regional
studies (Zhuo et al., 2016b). Those disadvantages restrict the full utilisation of satellite soil
moisture products for landslide monitoring application as discussed in Zhuo et al. (2019).
Another soil moisture data source relies on the state-of-the-art Land Surface Models (LSMs) such
as the Noah LSM (Ek et al., 2003) and the Community Land Model (CLM) (Oleson et al., 2010).
LSMs describe the interactions between the atmosphere and the land surface by simulating
exchanges of momentum, heat and water within the Earth system (Maheu et al., 2018). They are
capable of simulating the most important subsurface hydrological processes (e.g., soil moisture)
and can be integrated with the advanced Numerical Weather Prediction (NWP) system like WRF
(Weather Research and Forecasting) (Skamarock et al., 2008) for comprehensive soil moisture
estimations (i.e., through the surface energy balance, the surface layer stability and the water
balance equations) (Greve et al., 2013). NWP-based (i.e., with integrated LSM, thereafter) soil
moisture estimations have many advantages, for instance their spatial and temporal resolution can
be set discretionarily to fit different application requirements; their coverage is global, and the
estimated soil moisture data covers multiple soil layers (from the shallow surface layer to deep
root-zones); as well as a number of globally-covered data products can provide the necessary
boundary and initial conditions for running the models. Soil moisture estimated through such an
approach has been widely recognised and demonstrated in many studies, which cover a broad
range of applications from hydrological modelling (Srivastava et al., 2013a;Srivastava et al., 2015),
drought studies (Zaitchik et al., 2013), flood investigations (Leung and Qian, 2009), to regional
weather prediction (Stéfanon et al., 2014). Therefore, NWP-based soil moisture datasets could





provide valuable information for landslide applications. However, to our knowledge, relevant
research has never been carried out.
The aim of this study hence is to evaluate the usefulness of NWP modelled soil moisture for
landslide monitoring. Here the advanced WRF model (version 3.8) is adopted, because it offers
numerous physics options such as micro-physics, surface physics, atmospheric radiation physics,
and planetary boundary layer physics (Srivastava et al., 2015), and can integrate with a number of
LSM schemes, each varying in physical parameterisation complexities. So far there is limited
literature in comparing the soil moisture accuracy of different LSMs options in the WRF model.
Therefore, in this study, we select three of the WRF's most advanced LSM schemes (i.e., Noah,
Noah-Multiparameterization (Noah-MP), and CLM4) to compare their soil moisture performance
for landslide hazard assessment. Furthermore, since all the three schemes can provide multi-layer
soil moisture information, it is useful to include all those simulations for the comparison so that
the optimal depth of soil moisture could be determined for the landslide monitoring application.
The large physiographic variability, plus the abundance of the historical landslide events data,
makes Italy a good place for this research. Here an Italian region called Emilia Romagna is selected.
The study period covers 10 years from 2006 to 2015 to include a long-term record of landslide
events. In addition, because slope angle is a major factor controlling the stability of slope, it is
hence used in this study to divide the study area into several slope groups, so that a more accurate
threshold model could be built.
The description of the study area and the used datasets are included in Section 2. Methodologies
regarding the WRF model, the related LSM schemes and the adopted landslide threshold
evaluation approach are provided in Section 3. Section 4 shows the WRF soil moisture evaluation
results against the in-situ observations. Section 5 covers the comparison results of the WRF



modelled soil moisture products for landslide applications. The discussion and conclusion of the
study are included in Section 6.
**2.  Study Area and Datasets**
**2.1 Study Area**
The study area is in the Emilia Romagna Region, northern Italy (Figure 1). Its population density
is high. The region has high mountainous areas in the S-SW, and wide plain areas towards NE,
with a large elevation difference (i.e., 0 m to 2125 m) across 50 km distance from the north to the
south. The region has a mild Mediterranean climate with distinct wet and dry seasons (i.e., dry
season between May and October, and wet season between November and April). The study area
tends to be affected by landslide events easily, with one-fifth of the mountainous zone covered by
active or dormant landslide deposits. Rainfall is by far the primary triggering factor of landslides
in the region, followed by snow melting: shallow landslides are triggered by short but
exceptionally intense rainfall, while deep-seated landslides have a more complex response to
rainfall and are mainly caused by moderate but exceptionally prolonged (even up to 6 months)
periods of rainfalls (Segoni et al., 2015).
**2.2 Selection of The Landslide Events**
The landslides catalog is collected from the Emilia Romagna Geological Survey (Berti et al., 2012).
The information included in the catalog are: location, date of occurrence, the uncertainty of date,
landslide characteristics (dimensions, type, and material), triggering factors, damages, casualties,
and references. Unfortunately, many of the information are missing from the records in many cases.
In order to organise the data in a more systematic way so that only the relevant events are retained,
a two-step event selection procedure is initially carried out based on: 1) rainfall-induced only; and



2) high spatial-temporal accuracy (exact date and coordinates). Finally, a revision of the
information about the type of slope instabilities such as landslide/debris flow/rockfall and the
characteristics of the affected slope (natural or artificial) is also carried out over the selected
records (Valenzuela et al., 2018). The catalog period used in this study covers between 2006 and
2015, which is in accordance with the WRF' model run. After filtering the data records, only one-
fifths of them (i.e., 157 events) is retained. The retained events are shown as single circles in Figure
2, with slope information (calculated through the Digital Elevation Model (DEM) data) also
presented in the background.  It can be seen the spatial distribution of the occurred landslide events
is very heterogeneous, with nearly all of them occurred in the hilly regions. During the study period,
the regional landslide occurrence is mainly dominated by the spatial distribution of the weak earth
units and the critical rainfall periods.
**2.3 Datasets**
There is a total of 19 soil moisture stations available within the study area, however only one of
them at the San Pietro Capofiume (latitude 44° 39' 13.59", longitude 11° 37' 21.6") provides long-
term valid soil moisture retrievals (i.e., 2006 to 2017). We have checked the data from all the rest
of the stations, they are either absent (or have very big data gaps) or do not cover the research
period at all. Therefore, only the San Pietro Capofiume station is used for the WRF soil moisture
temporal evaluation. The soil moisture is measured from 10 cm to 180 cm deep in the soil at 5
depths, by the Time Domain Reflectometry (TDR) instrument. Data are recorded in the unit of
volumetric water content ($m^3/m^3$) and at daily timestep (Pistocchi et al., 2008). The data used in
this study is between 2006 and 2015. In order to select rainfall events for Year 2014 and 2015,
data from 200 tipping-bucket rain gauges are collected and analysed within the region.



To drive a NWP model like WRF for soil moisture simulations, several globally-coved data
products can be chosen for extracting the boundary and initial conditions information, for instance,
the European Centre for Medium-Range Weather Forecasts (ECMWF) reanalysis (ERA-Interim)
and the National Centre for Environmental Prediction (NCEP) reanalysis are two of the most
commonly used data products. It has been found by Srivastava et al. (2013b) that the ERA-Interim
datasets can provide better boundary conditions than the NCEP datasets for WRF hydro-
meteorological predictions in Europe, which is therefore adopted in this study to drive the WRF
model. The spatial resolution of the ERA-Interim is approximately 80 km. The data is available
from 1979 to present, containing 6-hourly gridded estimates of three-dimensional meteorological
variables, and 3-hourly estimates of a large number of surface parameters and other two-
dimensional fields. A comprehensive description of the ERA-Interim datasets can be found in (Dee
et al., 2011)
The Shuttle Radar Topography Mission (SRTM) 3 Arc-Second Global (~ 90m) DEM datasets is
downloaded and used as the basis for the slope degree calculations. SRTM DEM data has been
widely used for elevation related studies worldwide due to its high quality, near-global coverage,
and free availability (Berry et al., 2007).
**3. Methodologies**
**3.1 WRF Model and The Three Land Surface Model Schemes**
The WRF model is a next-generation, non-hydrostatic mesoscale NWP system designed for both
atmospheric research and operational forecasting applications (Skamarock et al., 2005). The model
is powerful enough in modelling a broad range of meteorological applications vary from tens of
metres to thousands of kilometres (NCAR, 2018). It has two dynamical solvers: the ARW



(Advanced Research WRF) core and the NMM (Nonhydrostatic Mesoscale Model) core. The
former has more complex dynamic and physics settings than the latter which only has limited
setting choices. Hence in this study WRF with ARW dynamic core (version 3.8) is used to perform
all the soil moisture simulations.
The main task of LSM within the WRF is to integrate information generated through the surface
layer scheme, the radiative forcing from the radiation scheme, the precipitation forcing from the
microphysics and convective schemes, and the land surface conditions to simulate the water and
energy fluxes (Ek et al., 2003). WRF provides several LSM options, among which three of them
are selected in this study as mentioned in the introduction: Noah, Noah-MP, and CLM4. Table 1
gives a simple comparison of the three models. The detailed description of the models is written
below in the order of increasing complexity in regards of the way they deal with thermal and
moisture fluxes in various layers of soil, and their vegetation, root and canopy effects
(Skamarock et al., 2008).

### 3.1.1 Noah


Noah is the most basic amongst the three selected LSMs. It is one of the 'second generation' LSMs
that relies on both soil and vegetation processes for water budgets and surface energy closures
(Wei et al., 2010). The model is capable of modelling soil and land surface temperature, snow
water equivalent, as well as the general water and energy fluxes. The model includes four soil
layers that reach a total depth of 2 m in which soil moisture is calculated. Its bulk layer of canopy
-snow-soil (i.e., lack the abilities in simulating photosynthetically active radiation (PAR),
vegetation temperature, correlated energy, and water, heat and carbon fluxes), 'leaky' bottom (i.e.,
drained water is removed immediately from the bottom of the soil column which can result in
much fewer memories of antecedent weather and climate fluctuations) and simple snow melt-thaw




dynamics are seen as the model's demerits (Wharton et al., 2013). Noah calculates the soil moisture
from the diffusive form of Richard's equation for each of the soil layer (Greve et al., 2013), and
the evapotranspiration from the Ball-Berry equation (considering both the water flow mechanism
within soil column and vegetation, as well as the physiology of photosynthesis (Wharton et al.,

211    2013)).

**3.1.2 Noah-MP**
Noah-MP (Niu et al., 2011) is an improved version of the Noah LSM, in the aspect of better
representations of terrestrial biophysical and hydrological processes. Major physical mechanism
improvements directly relevant to soil water simulations include: 1) introducing a more permeable
frozen soil by separating permeable and impermeable fractions (Cai, 2015), 2) adding an
unconfined aquifer immediately beneath the bottom of the soil column to allow the exchange of
water between them (Liang et al., 2003), and 3) the adoption of a TOPMODEL (TOPography
based hydrological MODEL)-based runoff scheme  (Niu et al., 2005) and a simple SIMGM
groundwater model (Niu et al., 2007) which are both important in improving the modelling of soil
hydrology. Noah-MP is unique compared with the other LSMs, as it is capable of generating
thousands of parameterisation schemes through the different combinations of "dynamic leaf,
canopy stomatal resistance, runoff and groundwater, a soil moisture factor controlling stomatal
resistance (the β factor), and six other processes" (Cai, 2015). The  scheme option used in the study
are: Ball-Berry scheme for canopy stomatal resistance, Monin-Obukhov scheme for surface layer
drag coefficient calculation, the Noah based soil moisture factor for stomatal resistance, the
TOPMODEL runoff with the SIMGM groundwater, the linear effect scheme for soil permeability,
the two-stream applied to vegetated fraction scheme for radiative transfer, the CLASS (Canadian





Land Surface Scheme) scheme for ground surface albedo option, and the Jordan (Jordan, 1991)
scheme for partitioning precipitation between snow and rain.

**231    3.1.3. CLM4**

CLM4 is developed by the National Center for Atmospheric Research (NCAR) to serve as the land
component of its Community Earth System Model (formerly known as the Community Climate
System Model) (Lawrence et al., 2012). It is a 'third generation' model that incorporates the
interactions of both nitrogen and carbon in the calculations of water and energy fluxes. Compared
with its previous versions, CLM4 (Oleson et al., 2008) has multiple enhancements relevant to soil
moisture computing. For instance, the model's soil moisture is estimated by adopting a improved
one-dimensional Richards equation (Zeng and Decker, 2009); the new version allows the dynamic
interchanges of soil water and groundwater through an improved definition of the soil column's
lower boundary condition  that is similar to the Noah-MP's (Niu et al., 2007). Furthermore, the
thermal and hydrologic properties of organic soil are included for the modelling which is based on
the method developed in (Lawrence and Slater, 2008). The total ground column is extended to 42
m depth, consisting 10 soil layers unevenly spaced between the top layer (0.0–1.8 cm) and the
bottom layers (229.6–380.2 cm), and 5 bedrock layers to the bottom of the ground column
(Lawrence et al., 2011). Soil moisture is estimated for each soil layer.

**246    3.2 WRF Model Parameterization**

The WRF model is centred over the Emilia Romagna Region with three nested domains (D1, D2,
D3 with the horizontal grid sizes of 45 km, 15 km, and 5 km, respectively), of which the innermost
domain (D3, with 88 x 52 grids (west-east and south-north, respectively)) is used in this study. A
two-way nesting scheme is adopted allowing information from the child domain to be fed back to



the parent domain. With atmospheric forcing, static inputs (e.g., soil and vegetation types), and
parameters, the WRF model needs to be spun-up to reach its equilibrium state before it can be used
(Cai et al., 2014;Cai, 2015). In this study, WRF is spun-up by running through the whole year of
2005. After spin-up, the WRF model for each of the selected LSM scheme is executed in daily
timestep from January 1, 2006, to December 31, 2015, using the ERA-Interim datasets.
The microphysics scheme plays a vital role in simulating accurate rainfall information which in
turn is important for modelling the accurate soil moisture variations. WRF V3.8 is supporting 23
microphysics options range from simple to more sophisticated mixed-phase physical options. In
this study, the WRF Single-Moment 6-class scheme is adopted which considers ice, snow and
graupel processes and is suitable for high-resolution applications (Zaidi and Gisen, 2018). The
physical options used in the WRF setup are Dudhia shortwave radiation (Dudhia, 1989) and Rapid
Radiative Transfer Model (RRTM) longwave radiation (Mlawer et al., 1997). Cumulus
parameterization is based on the Kain-Fritsch scheme (Kain, 2004) which is capable of
representing sub-grid scale features of the updraft and rain processes, and such a capability is
beneficial for real-time modelling (Gilliland and Rowe, 2007). The surface layer parameterization
is based on the Revised fifth-generation Pennsylvania State University–National Center for
Atmospheric Research Mesoscale Model (MM5) Monin-Obukhov scheme (Jiménez et al., 2012).
The Yonsei University scheme (Hong et al., 2006) is selected to calculate the planetary boundary
layer. The parameterization schemes used in the WRF modelling are shown in Table 2. The
datasets for land use and soil texture are available in the pre-processing package of WRF. In this
study, the land use categorisation is interpolated from the MODIS 21-category data classified by
the International Geosphere Biosphere Programme (IGBP). The soil texture data are based on the
Food and Agriculture Organization of the United Nations Global 5-minutes soil database.



**3.3 Translation of Observed and Simulated Soil Moisture Data to Common Soil Layers**
Since all soil moisture datasets have different soil depths, it is difficult for a direct comparison.
The Noah and Noah-MP models include four soil layers, centred at 5, 25, 70, and 150 cm,
respectively. Whereas CLM4 model has 10 soil layers, centered at 0.9, 3.2, 6.85, 12.85, 22.8, 39.2,
66.2, 110.65, 183.95, 304.9 cm, respectively. Moreover, the in-situ sensor measures soil moisture
centred at 10, 25, 70, 135, and 180 cm. In order to tackle the inconsistency issue of soil depths, the
simple linear interpolation approach described in Zhuo et al. (2015b) is applied in this study, and
a benchmark of soil layer centred at 10, 25, 70 and 150 cm is adopted.
**3.4 Soil Moisture Thresholds Build Up and Evaluations**
To build and evaluate the soil moisture thresholds for landslides forecasting, all datasets have been
grouped into two portions: 2006-2013 for the establishment of thresholds, and 2014-2015 for the
evaluation. The determination of soil moisture thresholds is based on determining the most suitable
soil moisture triggering level for landslides occurrence by trying a range of exceedance
probabilities (percentiles). For example, a 10% exceedance probability is calculated by
determining the 10% percentile result of the soil moisture datasets that is related to the occurred
landslides. The exceedance probability method is commonly utilised in landslide early warning
studies for calculating the rainfall-thresholds, which is therefore adopted here to examine its
performance for soil moisture threshold calculations.
To carry out the threshold evaluation, 45 rainfall events (during 2014-2015) are selected for the
purpose. The rainfall events are separated based on at least one-day of dry period (i.e., a period
without rainfall) (Dai et al., 2014;Dai et al., 2015;Dai et al., 2016). The rainfall data from each rain
gauge station is firstly combined using the Thiessen Polygon method, and with visual analysis, the



45 events are then finally selected. The information about the selected rainfall events can be found
in Section 5. The threshold evaluation is based on the statistical approach described in Gariano et
al. (2015) and Zhuo et al. (2019), where soil moisture threshold can be treated as a binary classifier
of the soil moisture conditions that are likely or unlikely to cause landslide events. With this
hypothesis, the likelihood of a landslide event can either be *true* (*T*) or *false* (*F*), and the threshold
forecasting can either be *positive* (*P*) or *negative* (*N*). The combinations of those four conditions
can lead to four statistical outcomes (Figure 3a) that are: *true positive* (*TP*), *true negative* (*TN*),
*false positive* (*FP*), and *false negative* (*FN*) (Wilks, 2011). The detailed description of each
outcome is covered in Zhuo et al. (2019). Using the four outcomes, two statistical scores can be
determined.
The Hit Rate (*HR*), which is the rate of the events that are correctly forecasted. Its formula is:
$HR = \frac{TP}{TP+FN}$                                                                   (1)
in the range of 0 and 1, with the best result as 1.
The False Alarm Rate *(FAR)*, which is the rate of false alarms when the event did not occur. Its
formula is:
$FAR = \frac{FP}{FP+TN}$                                                                  (2)
in the range of 0 and 1, with the best result as 0.
For any soil moisture product, each threshold calculated for each of the slope degree group is
adopted to determine *T*, *F*, *P*, and *N*, respectively. Those values are finally integrated to find the
overall scores of *TP, FN, FP, TN*, *HR,* and *FAR*. The threshold performance is then judged via the
Receiver Operating Characteristic (ROC) analysis (Hosmer and Lemeshow, 1989;Fawcett, 2006).
As shown in Figure 3b, ROC curve is based on *HR* against *FAR,* and each point in the curve
represents a threshold scenario (i.e., selected exceedance probabilities). The optimal result (the red
point) can only be realised when the *HR* reaches 1 and the *FAR* reduces to 0. The closer the point





to the red point, the better the forecasting result is. To analyse and compare the forecasting
performance numerically, the Euclidean distances (*d*) for each scenario to the optimal point are
computed.
**4.   WRF Soil Moisture Analysis and Evaluations**
**4.1 Temporal Comparisons**
Although there is only one soil moisture sensor that provides long-term soil moisture data in the
study region, it is still useful to compare it with the WRF estimated soil moisture. Particularly, it
has been shown that soil moisture measured at a site location can reflect the temporal fluctuations
of soil moisture for its surrounding region, up to 500 km in radius (Entin et al., 2000). With the
WRF's relatively high-resolution of 5 km, the temporal comparison with the in-situ observations
should provide some meaningful results. In this study, we carry out a temporal comparison
between all the three WRF soil moisture products with the in-situ observations. The comparison
is implemented over the period from 2006 to 2015, and the WRF grid closest to the in-situ sensor
location is chosen. Figure 4 shows the comparison results at the four soil depths. The statistical
performance (correlation coefficient *r* and Root Mean Square Error *RMSE*) of the three LSM
schemes are summarised in Table 3. Based on the statistical results, Noah-MP surpasses other
schemes at most soil layers, except for layer 2 where CLM4 shows stronger correlation and layer
4 where Noah gives smaller *RMSE* error. For Noah-MP, the best correlation is observed at the
surface layer (0.809), followed by third (0.738), second (0.683) and fourth (0.498) layers; and
based on *RMSE,* the best performance is again observed at the surface layer and followed by
second, third and fourth layers in sequence (as 0.060, 0.070, 0.088, and 0.092 m$^3$/m$^3$, respectively).
From the temporal plots, it can be seen at all four soil layers, all three LSM schemes can produce



soil moisture's seasonal cycle very well with most upward and downward trends successfully
represented. However, both the Noah and the CLM4 overestimate the variability at the upper two
soil layers during almost the whole study period, and the situation is the worst for the Noah.
Comparatively, the Noah-MP can capture the wet soil moisture conditions very well especially at
the surface layer; and it is the only model of the three that is able to simulate the large soil drying
phenomenon close to the observations during the dry season, except for some extremely dry days.
Towards 70 cm depth, although Noah-MP is still able to capture most of the soil moisture
variabilities during the drying period, it significantly underestimates soil moisture values for most
wet days. Similar underestimation results can be observed for CLM4 and Noah during the wet
season at 70 cm; furthermore, both schemes are again not capable of reproducing the extremely
drying phenomenon and overestimate soil moisture for most of the dry season days. It is surprising
to see that at the deep soil layer (150 cm), all soil moisture products are underestimated, in
particular, the outputs from the CLM4 and the Noah-MP only show small fluctuations. However,
the soil moisture measurements from the in-situ sensor also get our attention as they show strange
fluctuations with numerous sudden drops and rise situations observed. The strange phenomenon
is not expected at such a deep soil layer (although groundwater capillary forces can increase the
soil moisture, its rate is normally very slow). One possible reason we suspect is due to sensor
failure in the deep zone. Overall for the Noah-MP, in addition to producing the highest correlation
coefficient and the lowest *RMSE*, its simulated soil moisture variations are the closest to the
observations. The better performance of the Noah-MP over the other two models agrees with the
results found in Cai et al. (2014) (note: the paper uses standalone models, which are not coupled
with WRF). Also, it has been discussed in Yang et al. (2011), the Noah MP presents a clear
improvement over the Noah in simulating soil moisture globally.





## 4.2 Spatial Comparisons


Figure 5, 6 shows the spatial distribution of soil moisture simulations (via the three WRF LSM
schemes) at the four soil layers on a typical day during the dry and the wet seasons, respectively.
It is clear to see on the dry season day, Noah gives the wettest soil moisture simulation amongst
the three schemes, followed by CLM4 and Noah-MP. The soil moisture spatial pattern of the three
schemes more or less agrees with each, that is with wetter soil condition found in the central (in
line with the location of the river mainstream) and South-West part of the study region and dryer
soil condition in the Southern boundary and East part of the study region. On the wet season day,
Noah again produces wetter soil moisture data than the other two schemes, and it shows a distinct
wet patch at the Southern boundary while both the Noah-MP and the CLM4's simulations indicate
that part as the driest of the whole region. The disagreement among the LSMs at the Southern
boundary could be due to the particularly high elevation (above 2000 m) and snow existence at
that region, and the three schemes use different theories to deal with such conditions. The
improvement in the Noah-MP and the CLM4 is mainly attributed to the better simulation of snow,
in particular, it has been found Noah-MP can better simulate the snowmelt phenomenon over the
other two schemes (Cai et al., 2014), because it has better representations of ground heat flux,
retention, percolation and refreezing of melted liquid water within the multilayer snowpack (Yang
et al., 2011). Furthermore, it can be seen Noah-MP has a clear spatial pattern of the soil moisture
in the region, that is with drier areas found near the river mainstream, and Southern boundary, and
wetter zones in the North and the South. On the contrary, Noah and CLM4 simulated soil moisture
show a relatively smaller difference spatially, especially for CLM4.
**5. The Assessment of WRF Soil Moisture Threshold for Landslide Monitoring**





This section is to assess if the spatial distribution of soil moisture can provide useful information
for landslide monitoring at the regional scale. Particularly, all three soil moisture products
simulated through the WRF model are used to derive threshold models, and the corresponding
landslide prediction performances are then compared statistically. Here the threshold is defined as
the crucial soil moisture condition above which landslides are likely to happen.
Among different factors for controlling the stability of slope, the slope angle is one of the most
critical ones. From the slope angle map in Figure 2, it can be seen the region has a clear spatial
pattern of high and low slope areas, with the majority of the high-slope areas (can be as steep as
around 40 degrees) located in the mountainous Southern part and the river valleys. Moreover, there
is an obvious causal relationship between the slope angle and the landslide occurrence, as all the
landslides happened during the study period are located in the high-slope region, with a particularly
high concentration around the central Southern part. The spatial distribution of the landslide events
is also in line with the overall geological characteristics of the region, i.e., the Southern part mainly
constitutes outcrop of sandstone rocks that make up the steep slopes and are covered by a thin
layer of permeable sandy soil, which are highly unstable (Zhuo et al., 2019). Therefore, instead of
only using one soil moisture threshold for the whole study area, it is useful to divide the region
into several slope groups so that within each group a threshold model is built. To derive soil
moisture threshold individually under different slope conditions, all data has been divided into
three groups based on the slope angle (0.4-1.86°; 1.87-9.61°; 9.52-40.43°; since no landslide events
are recorded under the 0-0.39° group, the group is not considered here), as results,  all groups have
equal coverage areas.
In order to find the optimal threshold so that there are least missing alarms (i.e., threshold is
overestimated) and false alarms (i.e., threshold is underestimated), we test out 17 different





exceedance probabilities from 1% to 50%. For each LSM scheme, the total number of threshold
models is 204, which is the resultant of different combinations of slope groups, soil layers, and
exceedance probability conditions. The calculated thresholds for all LSM schemes under three
slope groups are plotted in Figure 7. Overall there is a very clear trend between the slope angle
and the soil moisture threshold, that is with threshold becoming smaller for steeper areas. The
correlation is particularly evident at the upper three soil layers (i.e., the top 1 m depth of soil), with
only a few exceptions for Noah and CLM4 at the 1% and the 2% exceedance probabilities.  At the
deep soil layer centred at 150 cm, the soil moisture threshold difference between Slope Group
(S.G.) 2 and 3 becomes very small for all the three LSM schemes. This could be partially because
at the deep soil layer, the change of soil moisture is much smaller than at the surface layer, therefore
the soil moisture values for S.G. 2 and 3 could be too similar to differentiate. However, for milder
slopes (S.G. 1), the higher soil moisture triggering level always applies even down to the deepest
soil layer for all the three LSM schemes. It is also clear to see the difference of threshold values
amongst different slope groups largely depends on the number of landslide events considered, that
is with more events considered, the stronger the correlation  (e.g., 1% exceedance probability
means 99% of the events are included for the threshold modelling, whilst 50% exceedance
probability means half of the data are treated as outliers). The results confirm that wetter soil
indeed can trigger shallow landslides easier in milder slopes than in steeper slopes.
All the threshold models are then evaluated under the 45 selected rainfall events (Table 4) using
the ROC analysis. The period of the selected rainfall events is between 1 day and 18 days, and the
average rainfall intensity ranges from 5.05 mm/day to 24.69 mm/day. For each selected event, the
number of landslide event is also summarised in the table. The resultant Euclidean distances ($d$)
between each scenario of exceedance probability and the optimal point for ROC analysis are listed



in Table 5 for all three WRF LSM schemes at the tested exceedance probabilities. The best
performance (i.e., lowest $d$) in each column (i.e., each soil layer of an LSM scheme) is highlighted.
In addition, the $d$ results are also plotted in Figure 8 to give a better view of the overall trend
amongst different soil layers and LSM schemes. From the figure, for all three LSM schemes at all
four soil layers, there is an overall downward and then stabilised trend. Overall for Noah, the
simulated surface layer soil moisture provides better landslide monitoring performance than the
rest of the soil layers from 1% to 35% exceedance probabilities; and the scheme's worst
performance is observed at the third soil layer centred at 70 cm. The values of $d$ for Noah's second
and fourth layer are quite close to each other. For Noah-MP, the simulated surface layer soil
moisture gives the best performance amongst all four soil layers for most cases between the 1%
and 35% exceedance probability range; and the scheme's worst performance is observed at the
fourth layer. Unlike Noah, all four soil layers from the Noah-MP scheme provide distinct
performance amongst them (i.e., larger $d$ difference). For CLM4, the performance for the surface
layer is quite similar to the second layer's, and the differences amongst the four layers are small.
From the Table 5, it can be seen for Noah the most suitable exceedance probabilities (i.e., the
highlighted numbers) range between 35% to 50%; for Noah-MP they are between 30% and 50%;
and for CLM4 it stays at 40% for all four soil layers. For both Noah and Noah-MP, the best
performance is observed at the surface layer ($d = 0.392$ and $d = 0.369$, respectively), which is in
line with their correlation coefficient results against the in-situ observations (i.e., the best $r$ value
for both LSM schemes is found at the surface layer). Furthermore, the best performance for Noah
and Noah-MP follows a regular trend, that is the deeper the soil layer, the poorer the landslide
monitoring performance. For CLM4, the best performances show no distinct pattern amongst soil
layers (i.e., with the best performance found at the soil layer 3, followed by layer 2, 1, and 4). Of



all the LSM schemes and soil layers, the best performance is found for Noah-MP at the surface
layer with 30% exceedance probability (*d*=0.369). The ROC curve for the Noah-MP scheme at the
surface layer is shown in Figure 9. In the curve, each point represents a scenario with a selected
exceedance probability level. It is clear with various exceedance probabilities, *FAR* can be
decreased without sacrificing the *HR* score (e.g., 4% to 10% exceedance probabilities). At the
optimal point at the 30% exceedance probability, the best results for *HR* and *FAR* are observed as
0.769 and 0.289, respectively.
**6.  Discussions and Conclusion**
In this study, the usability of WRF modelled soil moisture for landslide monitoring has been
evaluated in the Emilia Romagna region based on the research duration between 2006 and 2015.
Specifically, four-layer soil moisture information simulated through the WRF's three most
advanced LSM schemes (i.e., Noah, Noah-MP and CLM4) are compared for the purpose. Through
the temporal comparison with the in-situ soil moisture observations, it has been found that all three
LSM schemes at all four soil layers can produce soil moisture's seasonal cycle very well. However,
only Noah-MP is able to simulate the large soil drying phenomenon close to the observations
during the drying season, and it also gives the highest correlation coefficient and the lowest *RMSE*
at most soil layers amongst the three LSM schemes. For landslide threshold build up, slope
information is useful in identifying threshold differences, with threshold becoming smaller for
steeper area. In other words, dryer soil indeed can trigger landslides in steeper slopes than in milder
slopes. The result is not surprising, as the slope angle is an importance element of influencing the
stabilities of earth materials. Further studies based on slope angle condition is then carried out. 17
various exceedance probably levels between 1% and 50% are adopted to find the optimal threshold
scenario. Through the ROC analysis of 612 threshold models, the best performance is obtained by



the Noah-MP at the surface soil layer with 30% exceedance probability. The outstanding
performance of the Noah-MP scheme at the surface layer is also in accordance with its high
correlation coefficient result found against the in-situ observations.
It should be noted that weighting factors are not considered in the evaluation of the threshold
models. Nevertheless, in real-life situations, weighting could play important roles during the final
decision making. As for instance, the damages resulted from a missing alarm event could be much
more devastating than a false alarm event, or vice versa, and the situation also varies in different
regions. Therefore, during operational applications, weighting factors should be considered.
Model-based soil moisture estimations could be affected by error accumulation issues, especially
in the real-time forecasting mode. A potential solution is to use data assimilation methodologies
to correct such errors by intaking soil moisture information from other data sources. Since in-situ
soil moisture sensors are only sparsely available in limited regions, soil moisture measured via
satellite remote sensing technologies could provide useful alternatives. Another issue is with the
landslide record data, since most of them are based on human experiences (e.g., through
newspapers, and victims), a lot of incidences could be unreported. Therefore, the conclusion made
here could be biased. One way of expanding the current landslide catalog can depend on automatic
landslide detection methods based on remote sensing images.
In summary, this study gives an overview of the soil moisture performance of three WRF LSM
schemes for landslide hazard assessment. We demonstrate that the surface soil moisture (centred
at 10 cm) simulated through the Noah-MP LSM scheme is useful in predicting landslide
occurrences in the Emelia Romagna region. The high hitting rate of 0.769 and the low false alarm
rate of 0.289 obtained in this study show such valuable soil moisture information could work in
addition to the rainfall data to provide an efficient landslide early warning system at the regional





scales. However, one must bear in mind that the results demonstrated in this study are only valid

for the selected region. In order to make a general conclusion, more researches are needed.

Particularly, a considerable number of catchments with a broad spectrum of climate and

environmental conditions will need to be investigated.

**Acknowledgement**

This study is supported by Resilient Economy and Society by Integrated SysTems modelling

(RESIST), Newton Fund via Natural Environment Research Council (NERC) and Economic and

Social Research Council (ESRC) (NE/N012143/1), and the National Natural Science Foundation

of China (No:4151101234).

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



**Table 1.** Comparison of Noah, Noah-MP, and CLM4.

|  | Noah | Noah-MP | CLM4 |
|---|---|---|---|
| Energy balance | Yes | Yes | Yes |
| Water balance | Yes | Yes | Yes |
| No. of soil layers | 4 | 4 | 10 |
| Depth of total soil column | 2.0 m | 2.0 m | 3.802 m |
| Model soil layer thickness | 0.1, 0.3, 0.6, 1.0 m | 0.1, 0.3, 0.6, 1.0 m | 0.018, 0.028, 0.045, 0.075, 0.124, 0.204, 0.336, 0.553, 0.913, 1.506 m |
| No. of vegetation layers | A single combined surface layer of vegetation and snow | Single layer | Single layer |
| Vegetation | Dominant vegetation type in one grid cell with prescribed LAI | Dominant vegetation type in one grid cell with dynamic LAI | Up to 10 vegetation types in one grid cell with prescribed LAI |
| No. of snow layers | A single combined surface layer of vegetation and snow | Up to three layers | Up to five layers |



**Table 2.** WRF parameterizations used in this study

|  | Settings/ Parameterizations | References |
| --- | --- | --- |
| Map projection | Lambert |  |
| Central point of domain | Latitude: 44.54; Longitude: 11.02 |  |
| Latitudinal grid length | 5 km |  |
| Longitudinal grid length | 5 km |  |
| Model output time step | Daily |  |
| Nesting | Two-way |  |
| Land surface model | Noah, Noah-MP, CLM |  |
| Simulation period | 1/1/2006 – 31/12/2015 |  |
| Spin-up period | 1/1/2005 – 31/12/2005 |  |
| Microphysics | New Thompson | (Thompson et al., 2008) |
| Shortwave radiation | Dudhia scheme | (Dudhia, 1989) |
| Longwave radiation | Rapid Radiative Transfer Model | (Mlawer et al., 1997) |
| Surface layer | Revised MM5 | (Jiménez et al., 2012;Chen and Dudhia, 2001) |
| Planetary boundary layer | Yonsei University method | (Hong et al., 2006) |
| Cumulus Parameterization | Kain-Fritsch (new Eta) scheme | (Kain, 2004) |





**Table 3.** Statistical summary of the WRF performance in simulating soil moisture for different soil layers, based on comparison with the in-situ observations.

| | R | | | | RMSE ($m^3/m^3$) | | | |
|---|---|---|---|---|---|---|---|---|
| | 0.10 m | 0.25 m | 0.70 m | 1.50 m | 0.1 m | 0.25 m | 0.70 m | 1.50 m |
| Noah | 0.728 | 0.645 | 0.660 | 0.430 | 0.123 | 0.125 | 0.141 | **0.055** |
| Noah-MP | **0.809** | 0.683 | **0.738** | **0.498** | **0.060** | **0.070** | **0.088** | 0.092 |
| CLM | 0.789 | **0.743** | 0.648 | 0.287 | 0.089 | 0.087 | 0.123 | 0.089 |





**Table 4.** Rainfall events information.

| Starting date | | | Ending date | | | Duration (days) | Rainfall intensity (mm/day) | Number of Landslide events |
|---|---|---|---|---|---|---|---|---|
| Year | Month | Day | Year | Month | Day | | | |
| 2014 | 1 | 13 | 2014 | 1 | 24 | 12 | 20.50 | 2 |
| 2014 | 1 | 28 | 2014 | 2 | 14 | 18 | 13.61 | 0 |
| 2014 | 2 | 26 | 2014 | 3 | 6 | 9 | 13.35 | 0 |
| 2014 | 3 | 22 | 2014 | 3 | 27 | 6 | 11.08 | 0 |
| 2014 | 4 | 4 | 2014 | 4 | 5 | 2 | 18.98 | 0 |
| 2014 | 4 | 27 | 2014 | 5 | 4 | 8 | 12.13 | 0 |
| 2014 | 5 | 26 | 2014 | 6 | 3 | 9 | 5.05 | 0 |
| 2014 | 6 | 14 | 2014 | 6 | 16 | 3 | 18.29 | 0 |
| 2014 | 6 | 25 | 2014 | 6 | 30 | 6 | 11.39 | 0 |
| 2014 | 7 | 7 | 2014 | 7 | 14 | 8 | 7.84 | 0 |
| 2014 | 7 | 21 | 2014 | 7 | 30 | 10 | 15.35 | 0 |
| 2014 | 8 | 31 | 2014 | 9 | 5 | 6 | 5.67 | 0 |
| 2014 | 9 | 10 | 2014 | 9 | 12 | 3 | 11.84 | 0 |
| 2014 | 9 | 19 | 2014 | 9 | 20 | 2 | 23.04 | 0 |
| 2014 | 10 | 1 | 2014 | 10 | 1 | 1 | 14.51 | 0 |
| 2014 | 10 | 10 | 2014 | 10 | 17 | 8 | 13.01 | 0 |
| 2014 | 11 | 4 | 2014 | 11 | 18 | 15 | 18.28 | 0 |
| 2014 | 11 | 25 | 2014 | 12 | 7 | 13 | 7.58 | 0 |
| 2014 | 12 | 13 | 2014 | 12 | 16 | 4 | 6.24 | 0 |
| 2015 | 1 | 16 | 2015 | 1 | 17 | 2 | 14.87 | 0 |
| 2015 | 1 | 21 | 2015 | 1 | 23 | 3 | 7.13 | 0 |
| 2015 | 1 | 29 | 2015 | 2 | 10 | 13 | 9.98 | 0 |
| 2015 | 2 | 13 | 2015 | 2 | 17 | 5 | 6.62 | 1 |
| 2015 | 2 | 21 | 2015 | 2 | 26 | 6 | 11.84 | 4 |
| 2015 | 3 | 3 | 2015 | 3 | 7 | 5 | 11.69 | 1 |
| 2015 | 3 | 15 | 2015 | 3 | 17 | 3 | 9.00 | 0 |
| 2015 | 3 | 21 | 2015 | 3 | 27 | 7 | 12.09 | 2 |
| 2015 | 4 | 3 | 2015 | 4 | 5 | 3 | 16.62 | 0 |
| 2015 | 4 | 17 | 2015 | 4 | 18 | 2 | 6.99 | 0 |
| 2015 | 4 | 26 | 2015 | 4 | 29 | 4 | 11.23 | 0 |
| 2015 | 5 | 15 | 2015 | 5 | 16 | 2 | 8.83 | 0 |
| 2015 | 5 | 20 | 2015 | 5 | 27 | 8 | 10.58 | 1 |
| 2015 | 6 | 8 | 2015 | 6 | 11 | 4 | 6.47 | 0 |
| 2015 | 6 | 16 | 2015 | 6 | 19 | 4 | 13.44 | 0 |
| 2015 | 6 | 23 | 2015 | 6 | 24 | 2 | 6.07 | 0 |
| 2015 | 7 | 22 | 2015 | 7 | 25 | 4 | 6.05 | 0 |
| 2015 | 8 | 9 | 2015 | 8 | 10 | 2 | 24.69 | 0 |
| 2015 | 8 | 15 | 2015 | 8 | 19 | 5 | 10.69 | 0 |
| 2015 | 8 | 23 | 2015 | 8 | 24 | 2 | 7.88 | 0 |
| 2015 | 9 | 13 | 2015 | 9 | 14 | 2 | 24.66 | 1 |
| 2015 | 9 | 23 | 2015 | 9 | 24 | 2 | 7.50 | 0 |
| 2015 | 10 | 1 | 2015 | 10 | 7 | 7 | 13.73 | 0 |
| 2015 | 10 | 10 | 2015 | 10 | 19 | 10 | 9.40 | 0 |
| 2015 | 10 | 27 | 2015 | 10 | 29 | 3 | 20.33 | 0 |
| 2015 | 11 | 21 | 2015 | 11 | 25 | 5 | 13.78 | 1 |





**Table 5.** Results of Euclidean distances (*d*) between individual points and the optimal point for ROC analysis are listed. The best performance (i.e., lowest *d*) for each column (i.e., each soil layer of an LSM scheme) is highlighted. The optimal performance of all is highlighted in red.

| e.p. (%). | Noah 10 cm | 25 cm | 70 cm | 150 cm | Noah-MP 10 cm | 25 cm | 70 cm | 150 cm | CLM4 10 cm | 25 cm | 70 cm | 150 cm |
|---|---|---|---|---|---|---|---|---|---|---|---|---|
| 1 | 0.942 | 0.971 | 0.962 | 0.947 | 0.857 | 0.937 | 0.897 | 0.963 | 0.942 | 0.939 | 0.978 | 0.975 |
| 2 | 0.906 | 0.945 | 0.963 | 0.923 | 0.854 | 0.912 | 0.883 | 0.959 | 0.923 | 0.922 | 0.959 | 0.952 |
| 3 | 0.889 | 0.924 | 0.961 | 0.915 | 0.849 | 0.855 | 0.838 | 0.952 | 0.870 | 0.874 | 0.940 | 0.947 |
| 4 | 0.884 | 0.898 | 0.946 | 0.914 | 0.838 | 0.814 | 0.829 | 0.924 | 0.831 | 0.843 | 0.925 | 0.947 |
| 5 | 0.860 | 0.875 | 0.924 | 0.896 | 0.820 | 0.793 | 0.812 | 0.908 | 0.791 | 0.822 | 0.915 | 0.921 |
| 6 | 0.835 | 0.854 | 0.910 | 0.874 | 0.803 | 0.785 | 0.800 | 0.905 | 0.770 | 0.817 | 0.911 | 0.909 |
| 7 | 0.827 | 0.861 | 0.902 | 0.858 | 0.777 | 0.767 | 0.791 | 0.889 | 0.753 | 0.801 | 0.902 | 0.900 |
| 8 | 0.816 | 0.849 | 0.889 | 0.851 | 0.745 | 0.765 | 0.782 | 0.876 | 0.745 | 0.785 | 0.902 | 0.910 |
| 9 | 0.790 | 0.827 | 0.878 | 0.834 | 0.706 | 0.732 | 0.766 | 0.871 | 0.742 | 0.777 | 0.864 | 0.904 |
| 10 | 0.762 | 0.811 | 0.863 | 0.825 | 0.672 | 0.702 | 0.747 | 0.862 | 0.738 | 0.767 | 0.835 | 0.887 |
| 15 | 0.615 | 0.741 | 0.839 | 0.763 | 0.560 | 0.629 | 0.716 | 0.835 | 0.702 | 0.700 | 0.729 | 0.790 |
| 20 | 0.485 | 0.627 | 0.779 | 0.652 | 0.515 | 0.571 | 0.624 | 0.774 | 0.570 | 0.602 | 0.594 | 0.650 |
| 25 | 0.432 | 0.544 | 0.728 | 0.512 | 0.403 | 0.465 | 0.574 | 0.736 | 0.509 | 0.522 | 0.471 | 0.509 |
| 30 | 0.437 | 0.495 | 0.643 | 0.451 | **0.369** | **0.375** | 0.544 | 0.679 | 0.475 | 0.477 | 0.447 | 0.469 |
| 35 | **0.392** | 0.446 | 0.592 | 0.436 | 0.390 | 0.404 | 0.411 | 0.498 | 0.441 | 0.435 | 0.428 | 0.430 |
| 40 | 0.500 | **0.407** | 0.531 | 0.416 | 0.439 | 0.385 | **0.382** | 0.436 | **0.406** | **0.405** | **0.398** | **0.410** |
| 50 | 0.552 | 0.425 | **0.404** | **0.411** | 0.489 | 0.417 | 0.416 | **0.429** | 0.437 | 0.435 | 0.408 | 0.437 |



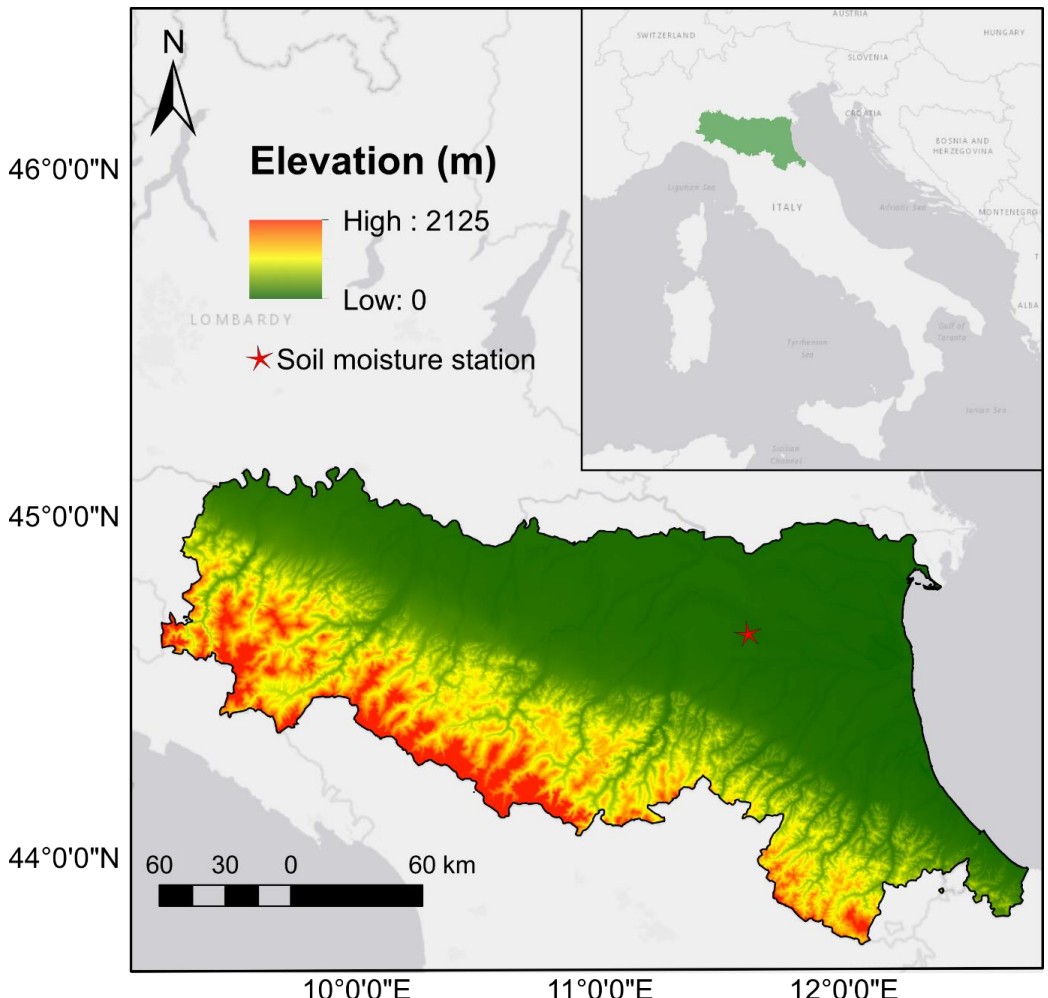

**Figure 1.** Location of the Emilia Romagna Region with elevation map and in-situ soil moisture station also shown.




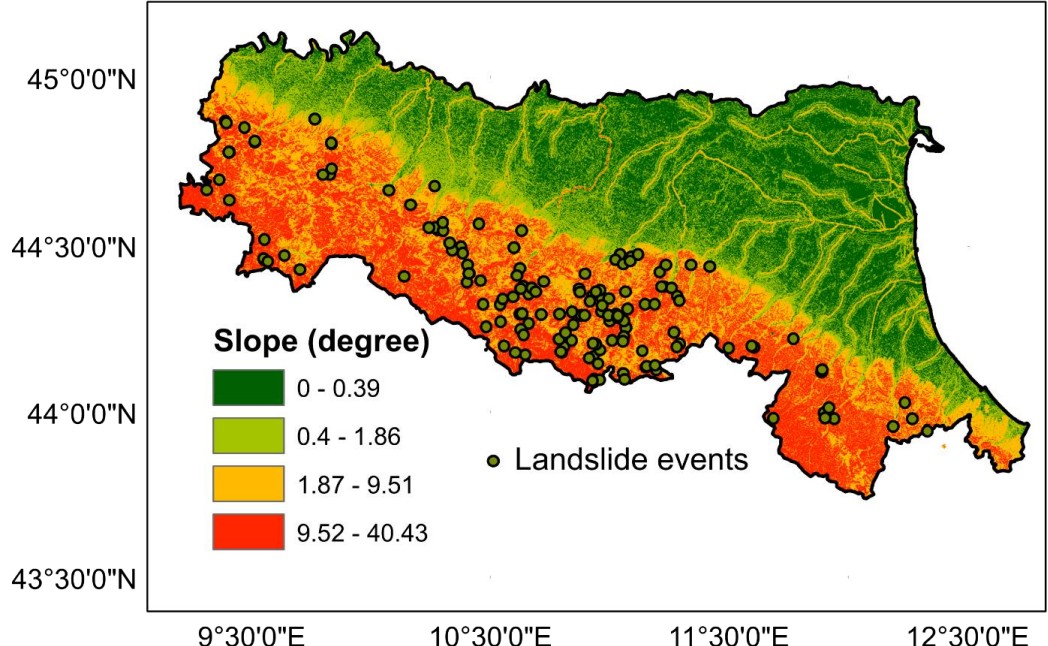

**Figure 2.** Landslide events with slope angle map.





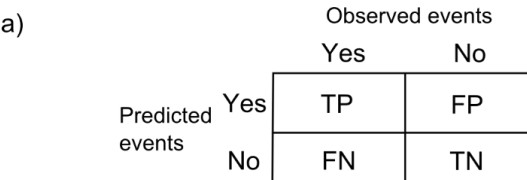

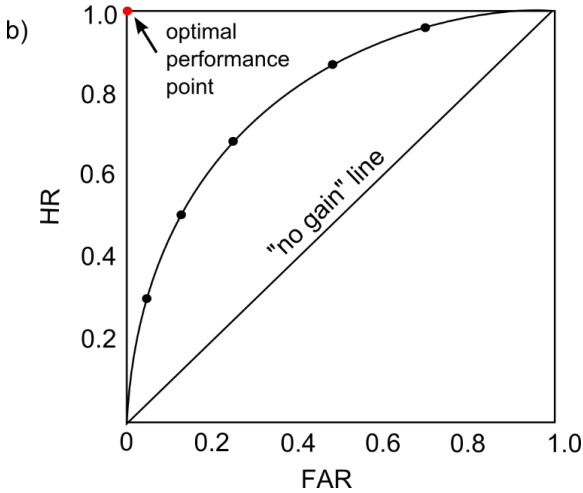

**Figure 3.** a) Contingency table illustrates the four possible outcomes of a binary classifier model: TP (True Positive), TN (True Negative), FP (False Positive), and FN (False Negative). b) ROC (Receiver Operating Characteristic) analysis with HR (Hitting Rate) against FAR (False Alarm Rate).





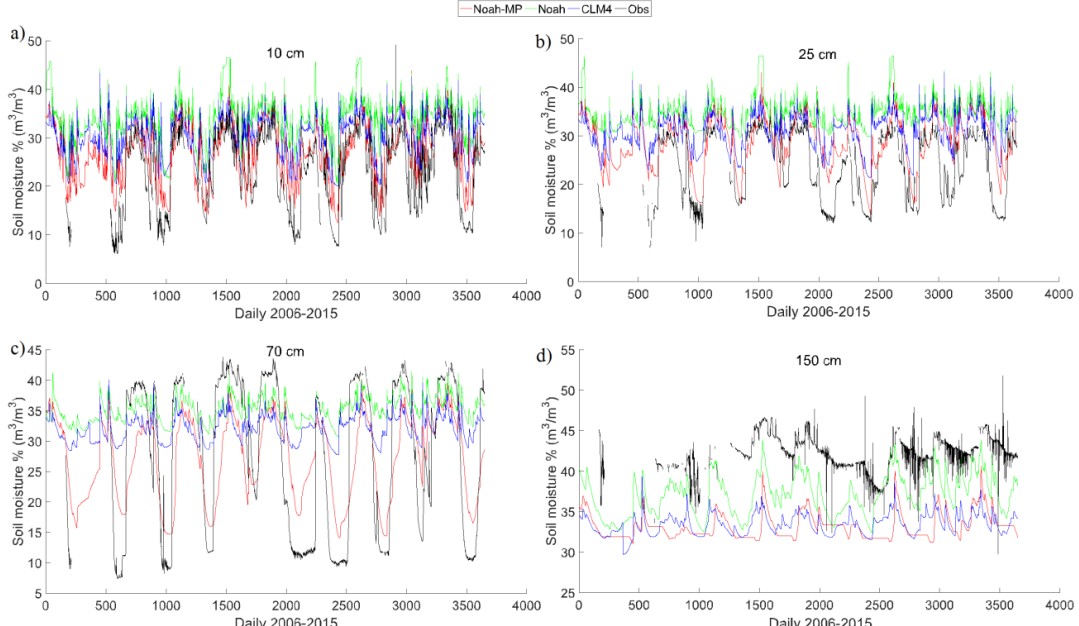

**Figure 4.** Soil moisture temporal variations of WRF simulations and in-situ observations for four soil layers at a) 10 cm; b) 25 cm; c) 70 cm; and d) 150 cm.




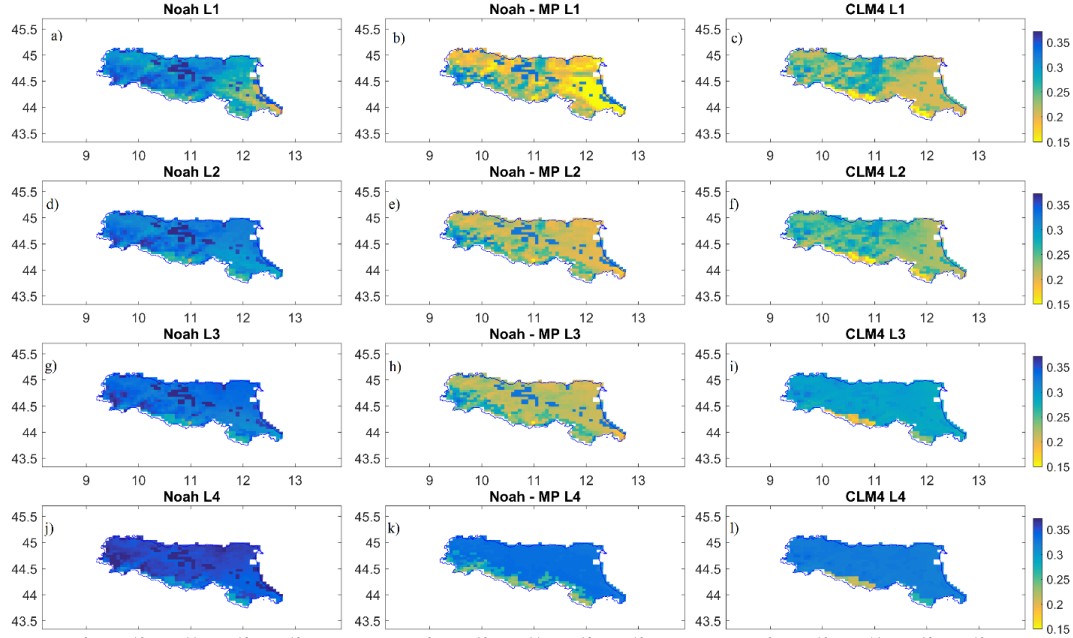

**Figure 5.** Spatial distribution of soil moisture at four soil layers (L1 = 10 cm; L2 = 25 cm; L3 = 70 cm; L4 = 150 cm) from WRF model simulations for Noah (a, d, g, j), Noah-MP (b, e, h, k), and CLM4 (c, f, i, l), on the August 1, 2015 (dry season).



Hydrology and
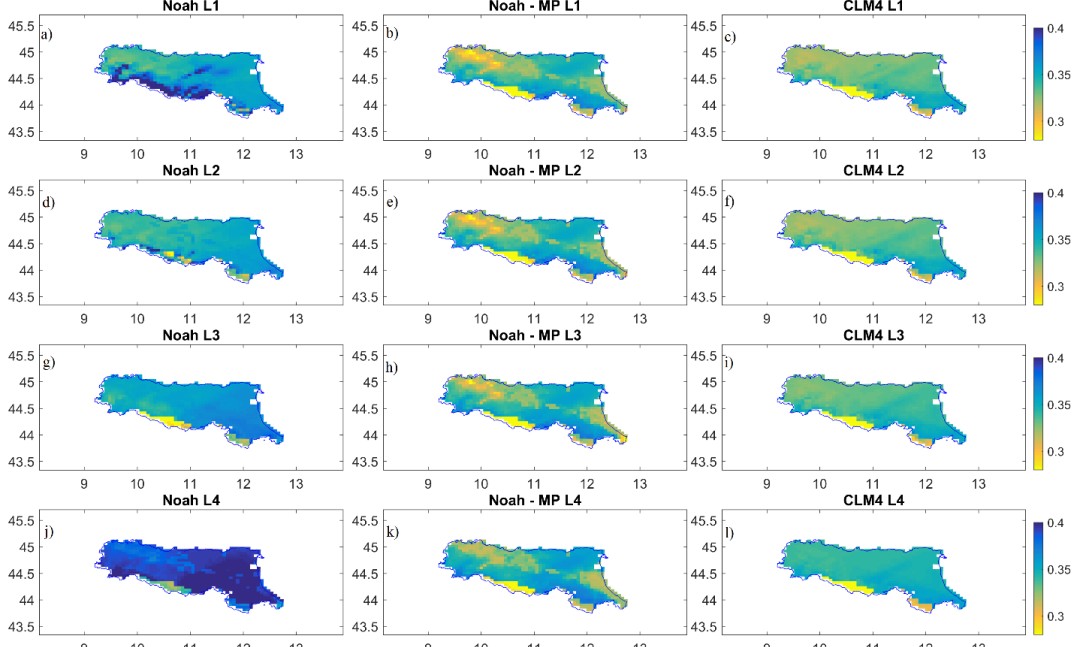

**Figure 6.** Spatial distribution of soil moisture at four soil layers (L1 = 10 cm; L2 = 25 cm; L3 = 70 cm; L4 = 150 cm) from WRF model simulations for Noah (a, d, g, j), Noah-MP (b, e, h, k), and CLM4 (c, f, i, l), on the February 1, 2015 (wet season).

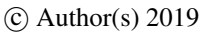



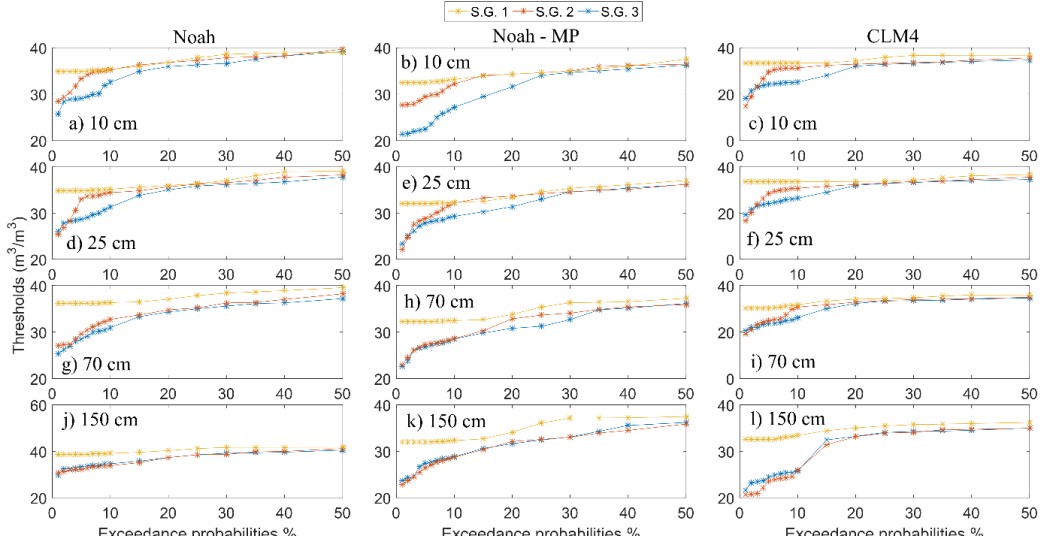

**Figure 7.** Threshold plots. For Noah (a, d, g, j), Noah-MP (b, e, h, k), and CLM4 (c, f, i, l) land surface schemes under three Slope angle Groups (S.G.) with S.G. 1 = 0.4-1.86º; S.G. 2 = 1.87-9.61º; S.G. 3 = 9.52-40.43º.





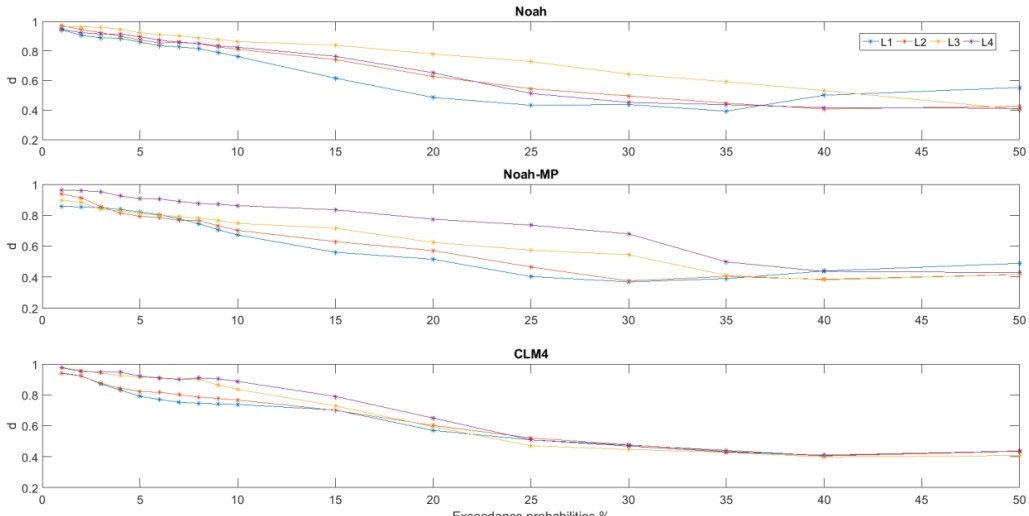

**Figure 8.** d-scores.



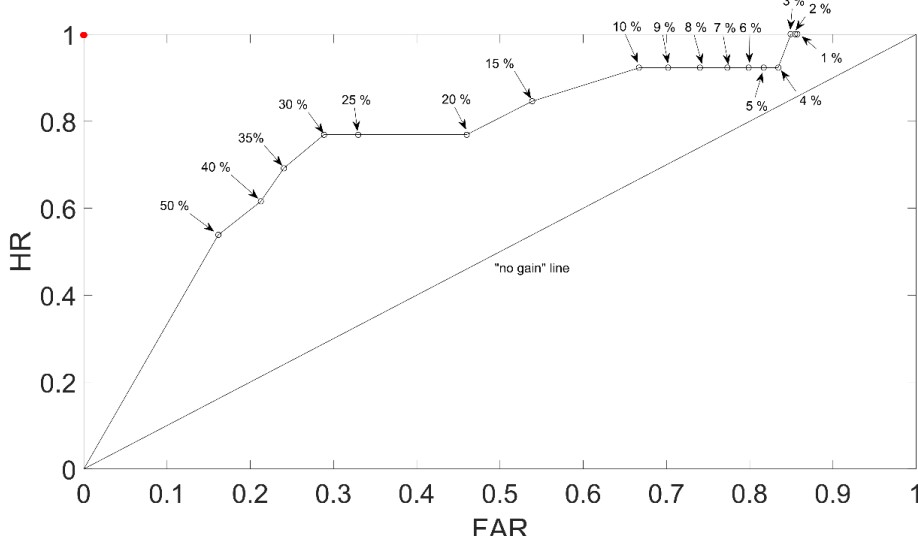

**Figure 9.** ROC curve for the calculated thresholds using different exceedance probability levels (for Noah-MP at the surface layer). The *no gain* line and the optimal performance point (the red point) are also presented.