# Peer review of "Assessment of Simulated Soil Moisture from WRF Noah, Noah-MP, and CLM Land Surface Schemes for Landslide Hazard Application"

_Hydrology and Earth System Sciences, 2019_

## Referee Comment (RC1) · Anonymous Referee #1 · 14 Apr 2019

**OVERVIEW**

The study investigates the use of modelled soil moisture data obtained from land surface modelling for the prediction of landslide occurrence. Specifically, three different versions of WRF model (three configurations for the land surface model scheme) are used for developing a soil moisture – based landslide threshold model in Emilia Romagna (Italy) in the period 2006-2015.

**GENERAL COMMENTS**

The paper is fairly well written and clear. The topic is surely of interest for the readership of "Hydrology and Earth System Sciences" journal. In recent years, the use of modelled and satellite soil moisture data are increasingly used for the prediction of landslides occurrence in space and time, and the study might represent an important contribution in this respect. However, in my opinion, some parts and aspects should be clarified before the publication.

I listed here the general comments also including their relevance:

1) MAJOR: The same authors have just published a similar paper on JSTARS over the same study area and using the same landslide catalogue for testing a soil moisture (and rainfall) threshold model. In the JSTARS paper, the authors have used satellite soil moisture data instead of modelled data. Firstly, the differences between the two studies should be clearly highlighted. Secondly, the comparison of the results obtained in the two studies should be carried out (the same 45 rainfall events are used for the ROC curve in the two studies). Is it better to use modelled or satellite soil moisture data for landslide prediction?

2) MODERATE: In the introduction, a brief description of limitations of satellite soil moisture data is given. However, I have found some errors: 1) microwave observations have not the problem of cloud cover, 2) with Sentinel-1 we have 1 km resolution / 3 days soil moisture observations (operationally available under the Copernicus Land Monitoring Service). Therefore, currently there is large potential in using satellite observations for landslide prediction, it should be clearly acknowledged.

3) MAJOR: It is not clear which soil moisture value is used. Initial soil moisture, final soil moisture at the end of rainfall event, maximum soil moisture, mean soil moisture?

It must be clarified. Moreover, it is well known that soil moisture is strictly related to rainfall, and I was wondering how accurate are the WRF simulated rainfall? A comparison between observed and simulated rainfall should be carried out to have a better understanding of the quality of WRF model in the study area.

4) MODERATE: It would be very relevant to perform a comparison with an approach based on rainfall threshold. Intensity-duration (or accumulated-duration) rainfall thresholds are largely used for landslide prediction. What is the accuracy of such an approach with respect to the one based on soil moisture proposed in the paper? This would add something new with respect to the JSTARS paper.

5) MODERATE: In the results, it is clearly shown that the soil moisture threshold percentiles are different for different slope angles. Then, it is not clear if the slope-dependence of soil moisture percentiles is used in the validation of the approach on the 45 rainfall events showing in Table 4 and Figure 9. It should be clarified.

I listed in the specific comments a number of corrections and changes that are needed.

**SPECIFIC COMMENTS (P: page, L: line or lines)**

P3, L60: Use of soil moisture for landslide prediction has been recently used. However, in Italy some studies using modelled soil moisture data have been published and I believe they should be mentioned (e.g., Ponziani et al., 2012, doi: 10.1007/s10346-011-0287-3; Ciabatta et al., 2016, doi: 10.1016/j.jhydrol.2016.02.007).

P4, L87: Spatial and temporal resolution of modelled data can not be set "discretionarily". It depends of many aspects, among them resolution of input observations and of maps used for the parameterization. Please revise.

P5, L112-113: Threshold of what? At this stage, it is not clear to what the authors refer. Please clarify.

P6, L127: 20-percent of mountainous area is covered by landslide. Is it correct? It seems to be overestimated.

P6, L129-130: Shallow landslides are not triggered by short and intense rainfall events only. Long and moderate rainfall events over saturated conditions may generate landslide events. Please revise.

P7, L144: Typo "WRF'"

P12, L252: Typo "spun-up", also at L253.

P12, L255: The ERA5 dataset is found to be better than ERA-Interim, also with a better spatial resolution. It should be used, at least for future studies.

P15, L328: 500 km radius seems to be too large. Please revise.

P16, L358-359: I believe that in situ soil moisture observations at deep layer are wrong, at least for some periods. Therefore, it should not be used for model evaluation.

P17, L365-385: The visualization of 2 soil moisture maps for 2 specific days has little sense to me. Better would be to perform a cross-correlation analysis in space and time to highlight the space-time agreement between the modelled datasets.

Figure 7: It is crowded, with too many lines. Please try to simplify.

Figure 8: Try to improve the visualization of the results in the figure.

P20, L453-454: It is quite unexpected that deeper soil moisture is less effective for landslide prediction. It should be explained, or at least discussed, this important aspect.

P22, L482: What is the "weighting factor" that should be considered?

**RECOMMENDATION**
On this basis, I found the topic of the paper relevant, and I suggest a moderate revision before the paper can be published on Hydrology and Earth System Sciences.

---

## Referee Comment (RC2) · Anonymous Referee #2 · 23 Apr 2019

I've read and carefully evaluated the manuscript and my opinion is that it needs major revisions before being published in HESS. Although the scientific significance is high, I think that the scientific quality is affected by several shortcomings. The manuscript is generally well written, the presentation quality is fair and could be improved in some parts. Please find hereafter my main concerns, divided among general and specific issues.

GENERAL ISSUES

1- The authors should clearly differentiate this work from their recently published "Evaluation of Remotely Sensed Soil Moisture for Landslide Hazard (IEEE-JSTARS 2019)"

and avoid that some introductory parts read very similar.

2- I am somehow concerned about the dataset used. Emilia Romagna is one of the Italian regions with the best environmental datasets, some of them also publicly available for free. Therefore, I wonder why a dataset period was chosen in which only a soil moisture station is available, and why a manuscript submitted in 2019 relies on datasets until only 2015 (all dataset used extend almost until present days). DEMs of the region are available at finer spatial resolutions (10m and 20m): why using a 90m resolution SRTM DEM? Lastly, the landslide dataset seems largely incomplete. Many works on the same test site (see following comment) used larger landslide dataset for the same time period. This shortcoming may let the readers question about the significance of results obtained.

3- The references of the manuscript are very biased. I think the authors cited almost their whole scientific production (e.g. at line 294: are three references from the same author necessary?), while they almost ignored what has been published on the same subject and in the same case of study. For instance: landslide characteristics could be better described making reference to some previous works (e.g. Bertolini et al., 2005; Rossi et al., 2010). Regional scale rainfall thresholds for the Emilia Romagna have been already published by Berti et al. (2012) using an I-D approach and by Martelloni et al. (2012) using antecedent rainfall. Regional scale landslide warning systems for the Emilia Romagna Region have been addressed in several papers (e.g. Lagomarsino et al., 2013; Segoni et al. 2018a). Lagomarsino et al. (2015) compared an I-D threshold model and an antecedent rainfall threshold model concluding that in Emilia Romagna the latter provides better performances, probably due to the complex hydrologic response of the hillslopes after rainfalls. Segoni et al., 2018b (already in your reference list) tested that the performances of the Emilia Romagna threshold system could be improved by integrating basin-scale soil moisture estimated by means of TOPKAPI model. I think all those antecedent works could be used to properly "set the stage" for your research. Berti, M., Martina, M. L. V., Franceschini, S.,

Pignone, S., Simoni, A., & Pizziolo, M. (2012). Probabilistic rainfall thresholds for landslide occurrence using a Bayesian approach. Journal of Geophysical Research: Earth Surface, 117(F4). Bertolini, G., Guida, M. & Pizziolo, M. Landslides (2005) 2: 302. https://doi.org/10.1007/s10346-005-0020-1 Segoni, S., Rosi, A., Fanti, R., Gallucci, A., Monni, A., & Casagli, N. (2018). A Regional-Scale Landslide Warning System Based on 20 Years of Operational Experience. Water, 10(10), 1297. Lagomarsino, D., Segoni, S., Fanti, R., & Catani, F. (2013). Updating and tuning a regional-scale landslide early warning system. Landslides, 10(1), 91-97. Martelloni, G., Segoni, S., Fanti, R., & Catani, F. (2012). Rainfall thresholds for the forecasting of landslide occurrence at regional scale. Landslides, 9(4), 485-495. Lagomarsino, D., Segoni, S., Rosi, A., Rossi, G., Battistini, A., Catani, F., & Casagli, N. (2015). Quantitative comparison between two different methodologies to define rainfall thresholds for landslide forecasting. Natural Hazards and Earth System Sciences, 15(10), 2413-2423. Rossi, M., Witt, A., Guzzetti, F., Malamud, B. D., & Peruccacci, S. (2010). Analysis of historical landslide time series in the Emilia‐Romagna region, northern Italy. Earth Surface Processes and Landforms, 35(10), 1123-1137.

4- At this scale of analysis, the attempt to relate modeled soil moisture to a single instrumented site is a too big stretch in my opinion. Please, consider also that the sensor is located in a completely different setting (wide alluvial plain) than the territory typically affected by landslides (hills and mountains). I think the trends of soil moisture could be largely unrelated (as also the authors stated at line 61) and the example at line 328 (500km radius) would not hold in a case study characterized by many differences and peculiarities like Emilia Romagna. The authors could maybe cite other authors that attempted to establish empirical correlations of hydrological variables in Emilia Romagna (Segoni et al., 2018 with soil moisture; Martelloni et al., 2013, with snowpack thickness), however they calibrated the relationships over smaller territorial units, not over the whole region. I think these works could be used to partially defend the approach used in the manuscript, but I don't think they can completely clear the feeling that just one single instrument for the whole region is insufficient.

[Figure]

5- The authors should be very careful in providing unbiased, objective and humble points of view. The feeling is that in some parts of the manuscript they are overreaching when describing the results obtained (e.g. "outstanding" at line 479). Indeed, in my opinion the results are questionable. Beside the issue of using an instrument located in the alluvial plain to model landslide occurrence in very different climatic, hydrologic and geo-morphologic settings, there is a clear problem of result evaluation: most of the results are presented as graphics where it is difficult to ascertain the goodness of the model fitting because a long dataset is compressed in a small figure and also a qualitative evaluation is hard (sections 4.1. and 4.2). Some quantitative validation is mandatory to better evaluate the results. We need to know the differences, how big they are, where/when are located and why they are present. Also, about abstract, results and discussion: I don't think WRF modelled soil moisture has been properly evaluated for landslide monitoring purposes (line 464-465). This work in my opinion can be considered a preliminary attempt towards that direction, but to reach the goal more and better data are needed, together with a more thorough and quantitative evaluation of the results. I suggest that the authors rephrase their statements.

SPECIFIC ISSUES

18. "landslide threshold model" is a very generic term. Please, be more specific.

40-42. Please adjust this sentence and provide more references if necessary. Caine was the first to establish an I-D threshold, but to my knowledge that threshold has never been used for a warning system. In addition, national scale landslide warning systems are not so common and not so many examples of prototypal or operation applications exist in the literature (e.g. Krogly et al., 2018; Rosi et al. 2016; Auflic et al., 2016). Indeed, threshold-based landslide warning systems are usually established for smaller areas (e.g. basins or regions or small alert zones), see e.g. Devoli et al. (2018), Baum and Godt (2010), Mathew et al. (2014) . . .

149-150. "weak earth units" is unclear. Please, rephrase.
237 "an improved"

279-280. Not clear, please rephrase.

303-304. I think the concepts of TP/TN/FP/FN are quite established, no need to make reference to other works.

313. Maybe I'm mistaken, but I don't think at this point the slope degree groups have been presented yet.

342. Please rephrase: "very well" cannot be used (see also general comments).

356-359. So, you are saying that the dataset has a bad quality? Maybe the dataset needs to be smoothed?

370. Please, revise the English.

378. Actually, in my opinion you don't have a reliable benchmark, so you can only say that the models provided different results, you cannot say which one provided the best results.

387. This part is very important, but you did not introduce it appropriately. This is strange, because in the introduction you cited many relevant papers (Glade et al., 2000; Brocca et al., 2008; Segoni et al., 2018; Bogaard and Greco 2018). Maybe you should spend a few words saying that previous works demonstrated that in complex geomorphologic settings (as Emilia Romagna) a rainfall threshold approach is too simple and more hydrologically-driven approaches need to be established.

396. I disagree. The relationship between the slope angle and the landslide triggering is not so straightforward and it depends on the landslide typology. E.g. many slow earth flows (which are abundant in Emilia Romagna) can occur also on very low slope gradients.

407. This choice is questionable. The landslide susceptibility does not necessarily have to be equally represented in the territory.

422-426. This is quite obvious, I wouldn't devote so much space to this.

427. Why are you mentioning shallow landslides? Until now, I figured out that you are trying to model landslides in general.

430-431. Please, rephrase this sentence.

453-454. Theoretically, the conditions at the sliding surface should be the ones with the best performances. Therefore, either you have very shallow landslides, or your results are not good at assessing the real soil moisture conditions that are actually triggering/predisposing landslide initiation.

475-476. Please rephrase.

479. Do not use "outstanding".

481. I failed to quickly check the correlation coefficient. Please, clearly write where it can be found by the readers.

499. To my experience, the applicability of a similar system in a warning system would be very limited because it would have a poor spatial resolution: warning would be issued over the whole region, thus having limited actual applicability.

628. Please, check.

JEMEC AUFLIČ M, ŠINIGOJ J, KRIVIC M, PODBOJ M, PETERNEL T, KOMAC M, 2016. Landslide prediction system for rainfall induced landslides in Slovenia (Masprem). GEOLOGIJA 59/2, 259-271 Baum RL, Godt JW (2010). Early warning of rainfall-induced shallow landslides and debris flows in the USA. Landslides, 7(3):259–272. Krøgli, I. K., Devoli, G., Colleuille, H., Boje, S., Sund, M., and Engen, I. K.: The Norwegian forecasting and warning service for rainfall- and snowmelt-induced landslides, Nat. Hazards Earth Syst. Sci., 18, 1427-1450, https://doi.org/10.5194/nhess-18-1427-2018, 2018 Martelloni, G., Segoni, S., Lagomarsino, D., Fanti, R., & Catani, F. (2013). Snow accumulation/melting model (SAMM) for integrated use in regional scale

landslide early warning systems. Hydrology and Earth System Sciences, 17(3), 1229-1240. Mathew, J., Babu, D. G., Kundu, S., Kumar, K. V., & Pant, C. C. (2014). Integrating intensity–duration-based rainfall threshold and antecedent rainfall-based probability estimate towards generating early warning for rainfall-induced landslides in parts of the Garhwal Himalaya, India. Landslides, 11(4), 575-588. Rosi, A., Peternel, T., Jemec-Auflič, M., Komac, M., Segoni, S., & Casagli, N. (2016). Rainfall thresholds for rainfall-induced landslides in Slovenia. Landslides, 13(6), 1571-1577.

---

## Author Comment (AC1) · 10 Jun 2019

**Replies to Reviewer 1**

**OVERVIEW**

The study investigates the use of modelled soil moisture data obtained from land surface modelling for the prediction of landslide occurrence. Specifically, three different versions of WRF model (three configurations for the land surface model scheme) are used for developing a soil moisture – based landslide threshold model in Emilia Ro- magna (Italy) in the period 2006-2015.

**GENERAL COMMENTS**

The paper is fairly well written and clear. The topic is surely of interest for the readership of "Hydrology and Earth System Sciences" journal. In recent years, the use of modelled and satellite soil moisture data are increasingly used for the prediction of landslides occurrence in space and time, and the study might represent an important contribution in this respect. However, in my opinion, some parts and aspects should be clarified before the publication.

I listed here the general comments also including their relevance:

1) MAJOR: The same authors have just published a similar paper on JSTARS over the same study area and using the same landslide catalogue for testing a soil moisture (and rainfall) threshold model. In the JSTARS paper, the authors have used satellite soil moisture data instead of modelled data. Firstly, the differences between the two studies should be clearly highlighted. Secondly, the comparison of the results obtained in the two studies should be carried out (the same 45 rainfall events are used for the ROC curve in the two studies). Is it better to use modelled or satellite soil moisture data for landslide prediction?

Reply: Agreed.

Firstly, in the introduction, some parts will be removed to reduce the similarity from the previous paper on explaining the existing research gaps. The shortcomings of the previous study will be added in the updated manuscript, to clearly explain the necessity and novelty of this study (i.e., the need of using high spatial (both horizontally and vertically) and temporal resolution soil moisture products for landslide application).

Secondly, as suggested by the reviewer, the best Euclidean distance d in the previous study of using satellite soil moisture is found as 0.51, while this study shows the best performance can reach 0.37 (by Noah-MP model at 10 cm depth). So based on this comparison, the WRF modelled soil moisture can provide better landslide prediction performance than the satellite (i.e., ESA CCI soil moisture product). We will include this comparison result in the updated manuscript.

2) MODERATE: In the introduction, a brief description of limitations of satellite soil moisture data is given. However, I have found some errors: 1) microwave observations have not the problem of cloud cover, 2) with Sentinel-1 we have 1 km resolution / 3 days soil moisture observations (operationally available under the Copernicus Land Monitoring Service). Therefore, currently there is large potential in using satellite observations for landslide prediction, it should be clearly acknowledged.

Reply: Agreed. We will update the information for satellite soil moisture estimation, and include the discussion of using Sentinel-1 satellite soil moisture retrievals for landslide application (e.g., its availability only over the recent years is a limiting factor to build reliable thresholds).

3) MAJOR: It is not clear which soil moisture value is used. Initial soil moisture, final soil moisture at the end of rainfall event, maximum soil moisture, mean soil moisture? It must be clarified. Moreover, it is well known that soil moisture is strictly related to rainfall, and I was wondering how accurate are the WRF simulated rainfall? A comparison between observed and simulated rainfall should be carried out to have a better understanding of the quality of WRF model in the study area.

Reply: In this study, the daily mean soil moisture is used. The reason for not using the antecedent soil moisture condition plus rainfall data on the day is because the purpose of this study is to explore the relationship between different WRF simulated soil moisture and landslides solely. In general, soil moisture is a predisposing factor for slope instability, while rainfall is the triggering factor. The same rainfall may trigger or not a landslide depending on the soil moisture content at the time of the rainfall event. The mean soil moisture on the day of the landslide implicitly account for both the initial soil moisture and the effective rainfall absorbed by the ground, and can be a robust indicator of the hydrological condition of the slope.

We thank the reviewer for suggesting on evaluating WRF model through its rainfall performance, and we will add this work as suggested in the updated manuscript. Currently, the WRF estimated soil moisture is only evaluated through the single point in-situ measurement, which has posed concerns from the Reviewer 2. The added work will provide a useful indicator of the accuracy of the WRF estimated soil moisture.

4) MODERATE: It would be very relevant to perform a comparison with an approach based on rainfall threshold. Intensity-duration (or accumulated-duration) rainfall thresholds are largely used for landslide prediction. What is the accuracy of such an approach with respect to the one based on soil moisture proposed in the paper? This would add something new with respect to the JSTARS paper.

Reply: The purpose of this study is to preliminary assess the relationship between the WRF modelled soil moisture products and the landslide events. Currently as aforementioned the mean soil moisture is adopted which account for both the initial soil moisture and the effective rainfall absorbed by the ground. Only after the evaluation of the soil moisture product, rainfall information can then be used together with the antecedent soil moisture information for forecasting and monitoring purposes. Due to page limits, those work will be in our future studies, and at that point, a comprehensive comparison of the new method and the traditional rainfall-only method will also be carried out.

5) MODERATE: In the results, it is clearly shown that the soil moisture threshold percentiles are different for different slope angles. Then, it is not clear if the slope dependence of soil moisture percentiles is used in the validation of the approach on the 45 rainfall events showing in Table 4 and Figure 9. It should be clarified.

Reply: The utilisation of different percentiles for different slope angle groups is applied in the validation. For the validation study, each threshold determined for each of the slope class is used for summarising the numbers of T, F, P, and N events. Those numbers are then combined

to determine the overall statistical indicators (i.e., HR, FAR, HK). We will clarify this in the updated manuscript.

I listed in the specific comments a number of corrections and changes that are needed.

**SPECIFIC COMMENTS (P: page, L: line or lines)**

P3, L60: Use of soil moisture for landslide prediction has been recently used. However, in Italy some studies using modelled soil moisture data have been published and I believe they should be mentioned (e.g., Ponziani et al., 2012, doi: 10.1007/s10346- 011-0287-3; Ciabatta et al., 2016, doi: 10.1016/j.jhydrol.2016.02.007).

Reply: Agreed. They will be added.

P4, L87: Spatial and temporal resolution of modelled data can not be set "discretionarily". It depends of many aspects, among them resolution of input observations and of maps used for the parameterization. Please revise.

Reply: Agreed. This will be revised.

P5, L112-113: Threshold of what? At this stage, it is not clear to what the authors refer. Please clarify.

Reply: Agreed. This will be revised.

P6, L127: 20-percent of mountainous area is covered by landslide. Is it correct? It seems to be overestimated.

Reply: Agreed. 20% is an estimation. The sentence will be revised.

P6, L129-130: Shallow landslides are not triggered by short and intense rainfall events only. Long and moderate rainfall events over saturated conditions may generate landslide events. Please revise.

Reply: Agreed. This will be revised.

P7, L144: Typo "WRF"'

Reply: This will be corrected.

P12, L252: Typo "spun-up", also at L253.

Reply: They will be corrected.

P12, L255: The ERA5 dataset is found to be better than ERA-Interim, also with a better spatial resolution. It should be used, at least for future studies.

Reply: Agreed. We will include this information in the discussion.

P15, L328: 500 km radius seems to be too large. Please revise.

Reply: Agreed. This will be revised. And the suggested WRF rainfall evaluation study can provide useful guidance of the models' soil moisture performance.

P16, L358-359: I believe that in situ soil moisture observations at deep layer are wrong, at least for some periods. Therefore, it should not be used for model evaluation.

Reply: We have included a discussion about the possible in-situ sensor failure at the deep layer (i.e., "However, the soil moisture measurements from the in-situ sensor also get our attention as they show strange fluctuations with numerous sudden drops and rise situations observed. The strange phenomenon is not expected at such a deep soil layer (although groundwater capillary forces can increase the soil moisture, its rate is normally very slow). One possible reason we suspect is due to sensor failure in the deep zone."). But we will clarify in the updated manuscript that 'therefore the deep-layer data should not be used for evaluation'.

P17, L365-385: The visualization of 2 soil moisture maps for 2 specific days has little sense to me. Better would be to perform a cross-correlation analysis in space and time to highlight the space-time agreement between the modelled datasets.

Reply: Agreed. The spatial maps will be deleted. The suggested rainfall evaluation study will provide an improved assessment of both the spatial and temporal soil moisture accuracy of the WRF model.

Figure 7: It is crowded, with too many lines. Please try to simplify.

Reply: Agreed. The figure will be improved.

Figure 8: Try to improve the visualization of the results in the figure.

Reply: Agreed. The figure will be improved.

P20, L453-454: It is quite unexpected that deeper soil moisture is less effective for landslide prediction. It should be explained, or at least discussed, this important aspect.

Reply: Agreed.

P22, L482: What is the "weighting factor" that should be considered?

Reply: Weighting factors can include both social and economic components, for instance it can include the cost of a disaster event (e.g., both short-term and long-term impacts), the cost of an evacuation (e.g., relocation cost, business shutdown), as well as the social impacts of both cases. More details will be included in the updated manuscript.

**RECOMMENDATION**

On this basis, I found the topic of the paper relevant, and I suggest a moderate revision before the paper can be published on Hydrology and Earth System Sciences.

---

## Author Comment (AC2) · 10 Jun 2019

**Replies to Reviewer 2**

I've read and carefully evaluated the manuscript and my opinion is that it needs major revisions before being published in HESS. Although the scientific significance is high, I think that the scientific quality is affected by several shortcomings. The manuscript is generally well written, the presentation quality is fair and could be improved in some parts. Please find hereafter my main concerns, divided among general and specific issues.

**GENERAL ISSUES**

1- The authors should clearly differentiate this work from their recently published "Evaluation of Remotely Sensed Soil Moisture for Landslide Hazard (IEEE-JSTARS 2019)" and avoid that some introductory parts read very similar.

Reply: Agreed. As the reply to Reviewer 1, in the introduction, some parts will be removed to reduce the similarity from the previous paper on explaining the existing research gaps. The shortcomings of the previous study will be added in the updated manuscript, to clearly explain the necessity and novelty of this study (i.e., the need of using high spatial (both horizontally and vertically) and temporal resolution soil moisture products for landslide application).

2- I am somehow concerned about the dataset used. Emilia Romagna is one of the Italian regions with the best environmental datasets, some of them also publicly available for free. Therefore, I wonder why a dataset period was chosen in which only a soil moisture station is available, and why a manuscript submitted in 2019 relies on datasets until only 2015 (all dataset used extend almost until present days). DEMs of the region are available at finer spatial resolutions (10m and 20m): why using a 90m resolution SRTM DEM? Lastly, the landslide dataset seems largely incomplete. Many works on the same test site (see following comment) used larger landslide dataset for the same time period. This shortcoming may let the readers question about the significance of results obtained.

Reply: The reason for choosing the 10-year period of 2006-2015 is because although there are 19 soil moisture sensors installed in the region, only the San Pietro Capofiume station can provide the longest continuous valid data (2006 to early 2017). We have checked all the other stations, and they are either absent from valid data (e.g., have very big data gaps) or do not have data at all. This will be further clarified in the updated version of the manuscript.

The reason for using the SRTM DEM data is because it covers most part of the world and is free to download, which makes the study applicable globally.

The landslide data is collected from the Emilia Romagna Geological Survey. There is a total of around 800 events occurred during the 10-year study period. The reason for choosing only one-fifth of the events (i.e., 157 events) is because we have to filter the data based on: 1) rainfall-induced only; 2) records with high spatial-temporal accuracy (exact date and coordinates); and 3) landslide only (other events such as debris flow, rockfall and collapse are excluded).

3- The references of the manuscript are very biased. I think the authors cited almost their whole scientific production (e.g. at line 294: are three references from the same author necessary?), while they almost ignored what has been published on the same subject and in the same case of study. For instance: landslide characteristics could be better described making reference to

some previous works (e.g. Bertolini et al., 2005; Rossi et al., 2010). Regional scale rainfall thresholds for the Emilia Romagna have been already published by Berti et al. (2012) using an I-D approach and by Martelloni et al. (2012) using antecedent rainfall. Regional scale landslide warning systems for the Emilia Romagna Region have been addressed in several papers (e.g. Lagomarsino et al., 2013; Segoni et al. 2018a). Lagomarsino et al. (2015) compared an I-D threshold model and an antecedent rainfall threshold model concluding that in Emilia Romagna the latter provides better performances, probably due to the complex hydrologic response of the hillslopes after rainfalls. Segoni et al., 2018b (already in your reference list) tested that the performances of the Emilia Romagna threshold system could be improved by integrating basin-scale soil moisture estimated by means of TOPKAPI model. I think all those antecedent works could be used to properly "set the stage" for your research. Berti, M., Martina, M. L. V., Franceschini, S., Pignone, S., Simoni, A., & Pizziolo, M. (2012). Probabilistic rainfall thresholds for landslide occurrence using a Bayesian approach. Journal of Geophysical Research: Earth Surface, 117(F4). Bertolini, G., Guida, M. & Pizziolo, M. Landslides (2005) 2: 302. https://doi.org/10.1007/s10346-005-0020-1 Segoni, S., Rosi, A., Fanti, R., Gallucci, A., Monni, A., & Casagli, N. (2018). A Regional-Scale Landslide Warning System Based on 20 Years of Operational Experience. Water, 10(10), 1297. Lagomarsino, D., Segoni, S., Fanti, R., & Catani, F. (2013). Updating and tuning a regional-scale landslide early warning system. Landslides, 10(1), 91-97. Martelloni, G., Segoni, S., Fanti, R., & Catani, F. (2012). Rainfall thresholds for the forecasting of landslide occurrence at regionalscale. Landslides, 9(4), 485-495. Lagomarsino, D., Segoni, S., Rosi, A., Rossi, G., Battistini, A., Catani, F., & Casagli, N. (2015). Quantitative comparison between two different methodologies to define rainfall thresholds for landslide forecasting. Natural Hazards and Earth System Sciences, 15(10), 2413-2423. Rossi, M., Witt, A., Guzzetti, F., Malamud, B. D., & Peruccacci, S. (2010). Analysis of historical landslide time series in the Emiliaâ˘ARˇ Romagna region, northern Italy. Earth Surface Processes and Landforms, 35(10), 1123-1137.

Reply: We thank the reviewer for the comments. We will modify the references in the updated manuscript, so that a wide range of relevant papers are cited.

4- At this scale of analysis, the attempt to relate modeled soil moisture to a single instrumented site is a too big stretch in my opinion. Please, consider also that the sensor is located in a completely different setting (wide alluvial plain) than the territory typically affected by landslides (hills and mountains). I think the trends of soil moisture could be largely unrelated (as also the authors stated at line 61) and the example at line 328 (500km radius) would not hold in a case study characterized by many differences and peculiarities like Emilia Romagna. The authors could maybe cite other authors that attempted to establish empirical correlations of hydrological variables in Emilia Romagna (Segoni et al., 2018 with soil moisture; Martelloni et al., 2013, with snowpack thickness), however they calibrated the relationships over smaller territorial units, not over the whole region. I think these works could be used to partially defend the approach used in the manuscript, but I don't think they can completely clear the feeling that just one single instrument for the whole region is insufficient.

Reply: Agreed. As suggested by Reviewer 1 ('it is well known that soil moisture is strictly related to rainfall, and I was wondering how accurate are the WRF simulated rainfall? A comparison between observed and simulated rainfall should be carried out to have a better understanding of the quality of WRF model in the study area'). We will add the rainfall

evaluation work in the updated manuscript, which will provide useful indicators for the accuracy (both spatially and temporally) of the WRF estimated soil moisture in the study region.

5- The authors should be very careful in providing unbiased, objective and humble points of view. The feeling is that in some parts of the manuscript they are overreaching when describing the results obtained (e.g. "outstanding" at line 479). Indeed, in my opinion the results are questionable. Beside the issue of using an instrument located in the alluvial plain to model landslide occurrence in very different climatic, hydrologic and geo-morphologic settings, there is a clear problem of result evaluation: most of the results are presented as graphics where it is difficult to ascertain the goodness of the model fitting because a long dataset is compressed in a small figure and also a qualitative evaluation is hard (sections 4.1. and 4.2). Some quantitative validation is mandatory to better evaluate the results. We need to know the differences, how big they are, where/when are located and why they are present. Also, about abstract, results and discussion: I don't think WRF modelled soil moisture has been properly evaluated for landslide monitoring purposes (line 464-465). This work in my opinion can be considered a preliminary attempt towards that direction, but to reach the goal more and better data are needed, together with a more thorough and quantitative evaluation of the results. I suggest that the authors rephrase their statements.

Reply: Agreed. The 'too optimistic' sentences will be rephrased throughout the paper.

**SPECIFIC ISSUES**

18. "landslide threshold model" is a very generic term. Please, be more specific.

Reply: Agreed. This will be modified.

40-42. Please adjust this sentence and provide more references if necessary. Caine was the first to establish an I-D threshold, but to my knowledge that threshold has never been used for a warning system. In addition, national scale landslide warning systems are not so common and not so many examples of prototypal or operation applications exist in the literature (e.g. Krogly et al., 2018; Rosi et al. 2016; Auflic et al., 2016). Indeed, threshold-based landslide warning systems are usually established for smaller areas (e.g. basins or regions or small alert zones), see e.g. Devoli et al. (2018), Baum and Godt (2010), Mathew et al. (2014) : : :

Reply: Agreed. The sentence will be adjusted.

149-150. "weak earth units" is unclear. Please, rephrase.

Reply: Agreed. This will be rephrased.

237 "an improved"

Reply: This will be corrected.

279-280. Not clear, please rephrase.

Reply: This will be rephrased.

303-304. I think the concepts of TP/TN/FP/FN are quite established, no need to make reference to other works.

Reply: Agreed. The reference will be removed.

313. Maybe I'm mistaken, but I don't think at this point the slope degree groups have been presented yet.

Reply: Agreed. The sentence will be modified.

342. Please rephrase: "very well" cannot be used (see also general comments).

Reply: Agreed. This will be rephrased.

356-359. So, you are saying that the dataset has a bad quality? Maybe the dataset needs to be smoothed?

Reply: We suspect it's due to the sensing failure at the deep-layer. As suggested by Reviewer 1, we will further clarify this issue.

370. Please, revise the English.

Reply: This will be revised.

378. Actually, in my opinion you don't have a reliable benchmark, so you can only say that the models provided different results, you cannot say which one provided the best results.

Reply: Agreed. This will be rephrased.

387. This part is very important, but you did not introduce it appropriately. This is strange, because in the introduction you cited many relevant papers (Glade et al., 2000; Brocca et al., 2008; Segoni et al., 2018; Bogaard and Greco 2018). Maybe you should spend a few words saying that previous works demonstrated that in complex geomorphologic settings (as Emilia Romagna) a rainfall threshold approach is too simple and more hydrologically-driven approaches need to be established.

Reply: Agreed. This sentence will be expanded as suggested by the reviewer.

396. I disagree. The relationship between the slope angle and the landslide triggering is not so straightforward and it depends on the landslide typology. E.g. many slow earth flows (which are abundant in Emilia Romagna) can occur also on very low slope gradients.

Reply: Agreed. This sentence will be rephrased.

407. This choice is questionable. The landslide susceptibility does not necessarily have to be equally represented in the territory.

Reply: There are different ways to group the slopes. In this study, three groups have been defined with similar sizes so that relatively reliable results could be achieved from the statistical point of view.

422-426. This is quite obvious, I wouldn't devote so much space to this.

Reply: Agreed. This part will be shortened.

427. Why are you mentioning shallow landslides? Until now, I figured out that you are trying to model landslides in general.

Reply: Agreed. This will be modified.

430-431. Please, rephrase this sentence.

Reply: Agreed. This will be rephrased.

453-454. Theoretically, the conditions at the sliding surface should be the ones with the best performances. Therefore, either you have very shallow landslides, or your results are not good at assessing the real soil moisture conditions that are actually triggering/predisposing landslide initiation.

Reply: One potential reason is that, the conditions at the sliding surface are important, but the soil moisture above it is also important (the loading should be heavier with more water in the upper layer soil). We will add more discussion about the results.

475-476. Please rephrase.

Reply: This will be rephrased.

479. Do not use "outstanding".

Reply: Agreed. This will be modified.

481. I failed to quickly check the correlation coefficient. Please, clearly write where it can be found by the readers.

Reply: Agreed. The correlation coefficient information will be added here.

499. To my experience, the applicability of a similar system in a warning system would be very limited because it would have a poor spatial resolution: warning would be issued over the whole region, thus having limited actual applicability.

Reply: Agreed. The WRF modelled soil moisture data could provide useful information for landslide studies, but for early warning system, a lot of additional information (e.g., particularly high resolution data) will be needed. The sentence will be rephrased.

628. Please, check.

JEMEC AUFLIˇC M, ŠINIGOJ J, KRIVIC M, PODBOJ M, PETERNEL T, KOMAC M, 2016. Landslide prediction system for rainfall induced landslides in Slovenia (Masprem). GEOLOGIJA 59/2, 259-271 Baum RL, Godt JW (2010). Early warning of rainfall-induced shallow landslides and debris flows in the USA. Landslides, 7(3):259– 272. Krøgli, I. K., Devoli, G., Colleuille, H., Boje, S., Sund, M., and Engen, I. K.: The Norwegian forecasting and warning service for rainfall- and snowmelt-induced landslides, Nat. Hazards Earth Syst. Sci., 18, 1427-1450, https://doi.org/10.5194/nhess- 18-1427-2018, 2018 Martelloni, G., Segoni, S., Lagomarsino, D., Fanti, R., & Catani, F. (2013). Snow accumulation/melting model (SAMM) for integrated use in regional scale landslide early warning systems. Hydrology and Earth System Sciences, 17(3), 1229- 1240. Mathew, J., Babu, D. G., Kundu, S., Kumar, K. V., & Pant, C. C. (2014). Integrating intensity–duration-based rainfall threshold and antecedent rainfall-based probability estimate towards generating early warning for rainfall-induced landslides in parts of the Garhwal Himalaya, India. Landslides, 11(4), 575-588. Rosi, A., Peternel, T., Jemec-Aufliˇc, M., Komac, M., Segoni, S., & Casagli, N. (2016). Rainfall thresholds for rainfall-induced landslides in Slovenia. Landslides, 13(6), 1571-1577.

Reply: The references will be checked.

---

## Author Response (AR2)

Dear authors, both the Referees acknowledge a significant improvement of the manuscript, but the two judgements are split. Regarding the issues, raised by the second Referee, about the results scarcely supported by the (few) data about soil moisture, I see his point (squeezing too much the data to get results), especially considering that the same data set has been already used in other papers of your group, as highlighted by Referee #1 (one of them is currently under consideration for possible publication in HESS).

So, to try to make this manuscript acceptable for publication in HESS, I invite to do as much as you can to demonstrate that the results you present, although poorly significant for the case under study, may represent an example of a possible methodology to improve currently adopted approaches to landslide hazard assessment. Also the other major concern by Referee #2, about the validation of the results, should be carefully considered in revising the manuscript, as well as other comments which were not addressed and to which a convincing rebuttal was not provided during the first round of review.

I look forward to receiving a newly revised version.

Best regards,

Roberto Greco

Reply: We thank the AE and reviewer's comments on further improving the manuscript. Regarding using the 'same datasets' for the other papers in our group, it is important to clarify that although the 3 papers use the same landslide event records, they all use different landslide triggering data/methods. This paper uses the WRF derived multi-layer soil moisture information to work out the landslide initiation threshold, which is the first attempt of its kind which has not been done by previous studies. Such a dataset is globally available with high spatial and temporal resolution, so it has the advantage over satellite (as compared in the last updated manuscript with our previous J-STARS paper) and rain gauge networks (as discussed in Introduction). We have added this clarification in the revised manuscript.

In addition, we have explored the spatial variation of soil moisture to demonstrate the soil moisture representation of a single soil moisture sensor over a large region (these new results are added in the latest version of the manuscript). Again, this is a novel approach in the landslide study. Based on the newly added results, although there is a significant elevation difference in the region, a single soil moisture sensor has high representation of a significant proportion of the study area as demonstrated by the correlation analysis. Although there is still a small proportion of the areas where the correlation is poor, this has prompt us to carry out a future study on the optimal design of soil moisture sensor network for landslide study. The need for such a stusy is based on the fact that there has been a lot of studies on the optimal rain gauge network design, but similar research on soil moisture sensor network design has been largely ignored by the research community. The WRF derived soil moisture data has a potential to provide the important soil moisture spatial information for an optimal design of soil moisture sensor network, which will be carried out via

Principal Component Analysis and cluster analysis. Therefore, the study described in the current
paper has paved a foundation for such a research.

We admit that this paper clearly cannot solve all the problems, but the results and methods are new
which deserve to be known by the community. We have attached the updated manuscript for your
consideration.

Yours Sincerely

Lu Zhuo, the lead author

                          **Replies to Reviewer 2**

MAJOR FLAWS

1-  The use of a single measuring station located in a completely different setting is still an
open issue in my opinion. The authors provided some justification (only available station)
and heavily modified the evaluation section. This is not enough. In my opinion a
publication in an important journal like HESS demands good data and good results. I
appreciated the methodology developed in your paper but you just do not have good
amount and quality of data, therefore you are borderline. You cannot use an excuse that
these are the best data you could get in this test site: limited data prevent to get reliable and
robust outcomes, thus endangering publication. In my first report (general issue #4), I
suggested an approach to overcome this situation, but you didn't implement it in the revised
manuscript. I briefly outline it again (this is just a quick sum-up, please develop it further
and add these concepts in different places in the manuscript, where appropriate). One
measuring station in the whole area is not sufficient to adequately calibrate the model (this
must be stated clearly) and can be used only to build a methodology and to obtain only
preliminary results. However, from this preliminary steps, relevant outcomes could be
obtained: other studies in the same test site established empirical correlations between
landslides and hydrogeological variables on smaller territorial units (see details in my
previous general issue #4). Thus, it could be inferred that the proposed methodology is
preliminary but it could be further refined (and better result could be obtained) if data from
a denser measuring network would be available.
2-  Validation. What I understand from the latest version of your paper is the following: You
model soil moisture and rainfall. You provide a spatial validation only for the modeled
rainfall. You use the soil moisture modeled across the whole region. I think this is not
correct. This issue is related to the previous one: a measuring station in the alluvial plain
cannot be used to calibrate a soil moisture model in mountains hundreds of kilometers
away. However, you did it, you got some results in trying to predict landslides, you need
to add that more measuring instruments would allow for a better calibration potentially
improving the overall results.

Reply:

We thank the reviewer for raising the soil moisture sensor representation problem. We have
added the following clafications in the updated manuscript:

------------------------------

For the WRF soil moisture evaluation, clearly the evaluation work based on a single soil
moisture sensor located in plain area is not sufficient to derive conclusions about the model's
performance over the whole study region. Therefore, the results are preliminary here. However,
in this study, by introducing the WRF spatial soil moisture information into the landslide
prediction model, the performance indeed has been improved in comparison with our previous
study using the satellite remote sensing soil moisture data (Zhuo et. al 2019). A similar concept
has been carried out by Segoni et al., (2018b), who implemented the soil moisture information
simulated from a hydrological model into a regional landslide early warning system with clear
improvements in false/ missing alarm performance. Although the results shown in this study
is preliminary and confined by the study area, the improved landslide prediction performance
is already obtained. Therefore, it is hoped with more densely soil moisture network data
available globally and further refinements of the method, the results could be improved further.

------------------------------

In addition, ideally, it will be useful if there is a dense in-situ soil moisture sensing network
covering the whole study area. In reality, that's not practical, so we have to rely on the spatial soil
moisture information by other means. So far, the soil moisture data with the best spatial and
temporal resolution is from the WRF model. A question is about how representative of a single
soil moisture sensor for the whole study area. We have carried out the correlation study of the
single sensor with the whole study region (added in the discussion section). The initial assumption
is that the soil moisture sensor can only represent its adjacent area, but the result was a surprise.
Based on the results, a single point can represent a significant proportion of the region. Admittedly,
there are some areas where the correlations are poor and further studies are needed to find out why
some areas are highly correlated whereas others are not. This has prompt us to do a future study
on the optimal soil moisture sensor network design for landside applications. Although there are
numerous studies on the rain gauge network design by the research community, the soil moisture
sensor network design has been largely ignored by the community. Therefore, this study has paved
a foundation for such research.

---
PREVIOUS COMMENTS NOT ADDRESSED

- Former general issue #4: see also my general comment. You "dodged" the problem by
validating rainfall instead of soil moisture. But in the manuscript you use soil moisture,
therefore the problem remains still open.

Reply: Please see the reply above.

- Former specific issue L40-42. I can0t find any trace of this. Maybe because you delated
relecant parts of the introduction?

Reply: We thank the reviewer for providing us with the detailed references on the rainfall
landslide prediction methods. Since this study is focused on the soil moisture landslide
application, after reading the introduction section several times, we decided to remove this part
to avoid potential confusion.

- Former specific issue Line 628. I suggested to check line 628, not to check the references.
Sorry for the misunderstanding (I wrote last comment and the reference list too close each
other).

Reply: The former specific issue Line 628 was not supposed to be there, which is now removed.

---
GENERAL COMMENTS

The validation of the rainfall does not seem to provide good results: a R value of 0.40 seems
rather low to me. You make reference to another work in the USA that gets similar values, but
I think that the standards in the international literature are higher than this.
Results, discussion and conclusion are not well separated. The results section includes some
interpretation of the results (usually more convenient in the discussion). If the
discussion/conclusion section becomes too long, maybe it would be better to split discussion
and conclusions.

Reply: We agree that the rainfall validation is not good in this case study. Rainfall is one of
the main drivers of soil moisture change, and it is logical to think soil moisture and rainfall are
highly linked. However, because rainfall temporal vairation is of high frequency while soil
moisture is of low frequency, they behave differently. The results illustrate that for landslide
study, it is better to use the WRF soil moisture data rather than its rainfall data. Clearly more
studies are needed to confirm this assumption. We have included this explanation in the
updated manuscript.

We agree with the reviewer that some of the interpretation parts in the result section should be
moved to the discussion section. Since we have further added some new results in the
discussion section. We have split the discussion and conclusion into two separate parts.

---
MINOR COMMENTS

Please, check the bi-directional correspondence between the references in the text and in the
reference list, as you ou modified the introduction and it seems to me that some discrepancies
exists (e.g. Zhuo et al 2016 is cited at line 80 but it is not in the reference list).

Reply: The discrepancies are caused by the Endnote reference software, which occurred in the
'tracked changes' version only (e.g., Zhuo et al 2016 has been removed in the 'updated'
version). In this submission, we will make sure the discrepancies dont happen again.

L78. The new added sentence needs some reference.

Reply: The relevant references have been added in the updated manuscript.

In the first part of section 4.1 (e.g. L365 or 366), please clarify that the comparison is carried
out at a single point: the one where the measuring station is located.

Reply: This has been clarified throughout the updated manuscript. "In this study, we carry out
a temporal comparison between all the three WRF soil moisture products with the in-situ
observations (at a single soil moisture measuring point in the plain area)"

L441. Which works? Please be more specific.

Reply: It refers to the works mentioned in the introduction section. We have specified this in
the updated manuscript, but the references are not repeated in this section. "As introduced at
the beginning of the paper, previous works (as discussed in the Introduction) have
demonstrated that in complex geomorphologic settings (e.g., in Emilia Romagna), a rainfall
threshold approach is too simple and more hydrologically driven approaches need to be
established."

L465-466. I don't agree with this reason. You are dealing with landslides, not with pure
statistics, therefore the statistical reliability of this approach is questionable. I think it is better
to state in the premises that your objective is to have equal coverage areas, consequently you
identified those class-break values.

Reply: As suggested by the reviewer, the sentence has been updated as "There are different
ways to group the slopes. In this study, in order to have equal coverage areas, we have
identified these class-break values."

L519. Many shallow instead of very shallow?

Reply: We mean very shallow, but to avoid confusion, we have changed the sentence to "Third,
the landslides occurred in the region are mainly in the top shallow soil layer".

L549. Which study? Please provide a reference.

Reply: The relevant reference has been added. "The WRF rainfall performance is found to be
similar to a study carried out over the central USA (Van Den Broeke et al., 2018)."

L560-568- I suggest deleting this whole part as it includes issues like civil protection
procedures, risk perception, risk management and it goes beyond the scopes of your work.

Reply: We agree. This part has been deleted.

L570 Which studies? Please provide references.

Reply: The relevant reference has been added. "Here, WRF is modelled based on the ERA-Interim datasets, however, it has been found in Albergel et al. (2018), the performance of using the ERA5 has surpassed the ERA-Interim."

L580 Here some reference are needed. I suggest using Nichol and Wong 2005 for remote sensing (feel free to add more examples). In addition, I suggest to add also the possibility to use internet news to detect all relevant landslides (all landslides with a relevant impact on society will be reported on internet news), also using automatic methods (Battisitni et al., 2013).

Reply: The suggested contents have been added. "Other ways of expanding the current landslide catalog can depend on automatic landslide detection methods based on remote sensing images (Nichol and Wong, 2005;Chen et al., 2018), internet news (as all landslides with a relevant impact on society will be reported on internet news), and automatic web data mining methods (Battistini et al., 2013;Goswami et al., 2018)"

L588. I would add: "However, the methodology could be replicated to derive site-specific calibrations of the approach proposed."

[revised manuscript text omitted]
 | Noah 25 cm | Noah 70 cm | Noah 150 cm | Noah-MP 10 cm | Noah-MP 25 cm | Noah-MP 70 cm | Noah-MP 150 cm | CLM4 10 cm | CLM4 25 cm | CLM4 70 cm | CLM4 150 cm |
|---|---|---|---|---|---|---|---|---|---|---|---|---|
| 1 | 0.942 | 0.971 | 0.962 | 0.947 | 0.857 | 0.937 | 0.897 | 0.963 | 0.942 | 0.939 | 0.978 | 0.975 |
| 2 | 0.906 | 0.945 | 0.963 | 0.923 | 0.854 | 0.912 | 0.883 | 0.959 | 0.923 | 0.922 | 0.959 | 0.952 |
| 3 | 0.889 | 0.924 | 0.961 | 0.915 | 0.849 | 0.855 | 0.838 | 0.952 | 0.870 | 0.874 | 0.940 | 0.947 |
| 4 | 0.884 | 0.898 | 0.946 | 0.914 | 0.838 | 0.814 | 0.829 | 0.924 | 0.831 | 0.843 | 0.925 | 0.947 |
| 5 | 0.860 | 0.875 | 0.924 | 0.896 | 0.820 | 0.793 | 0.812 | 0.908 | 0.791 | 0.822 | 0.915 | 0.921 |
| 6 | 0.835 | 0.854 | 0.910 | 0.874 | 0.803 | 0.785 | 0.800 | 0.905 | 0.770 | 0.817 | 0.911 | 0.909 |
| 7 | 0.827 | 0.861 | 0.902 | 0.858 | 0.777 | 0.767 | 0.791 | 0.889 | 0.753 | 0.801 | 0.902 | 0.900 |
| 8 | 0.816 | 0.849 | 0.889 | 0.851 | 0.745 | 0.765 | 0.782 | 0.876 | 0.745 | 0.785 | 0.902 | 0.910 |
| 9 | 0.790 | 0.827 | 0.878 | 0.834 | 0.706 | 0.732 | 0.766 | 0.871 | 0.742 | 0.777 | 0.864 | 0.904 |
| 10 | 0.762 | 0.811 | 0.863 | 0.825 | 0.672 | 0.702 | 0.747 | 0.862 | 0.738 | 0.767 | 0.835 | 0.887 |
| 15 | 0.615 | 0.741 | 0.839 | 0.763 | 0.560 | 0.629 | 0.716 | 0.835 | 0.702 | 0.700 | 0.729 | 0.790 |
| 20 | 0.485 | 0.627 | 0.779 | 0.652 | 0.515 | 0.571 | 0.624 | 0.774 | 0.570 | 0.602 | 0.594 | 0.650 |
| 25 | 0.432 | 0.544 | 0.728 | 0.512 | 0.403 | 0.465 | 0.574 | 0.736 | 0.509 | 0.522 | 0.471 | 0.509 |
| 30 | 0.437 | 0.495 | 0.643 | 0.451 | **0.369** | **0.375** | 0.544 | 0.679 | 0.475 | 0.477 | 0.447 | 0.469 |
| 35 | **0.392** | 0.446 | 0.592 | 0.436 | 0.390 | 0.404 | 0.411 | 0.498 | 0.441 | 0.435 | 0.428 | 0.430 |
| 40 | 0.500 | **0.407** | 0.531 | 0.416 | 0.439 | 0.385 | **0.382** | 0.436 | **0.406** | **0.405** | **0.398** | **0.410** |
| 50 | 0.552 | 0.425 | **0.404** | **0.411** | 0.489 | 0.417 | 0.416 | **0.429** | 0.437 | 0.435 | 0.408 | 0.437 |

[Figure]

**Figure 1.** Location of the Emilia Romagna Region with elevation map and in-situ soil moisture station also shown.

[Figure]

**Figure 2.** Landslide events with slope angle map.

[Figure]

[Figure]

**Figure 3.** a) Contingency table illustrates the four possible outcomes of a binary classifier model: TP (True Positive), TN (True Negative), FP (False Positive), and FN (False Negative). b) ROC (Receiver Operating Characteristic) analysis with HR (Hitting Rate) against FAR (False Alarm Rate).

[Figure]

**Figure 4.** Soil moisture temporal variations of WRF simulations and in-situ observations for four soil layers at a) 10 cm; b) 25 cm; c) 70 cm; and d) 150 cm.

[Figure]

**Figure 5.** Rainfall evaluation: spatial distribution of the correlation coefficient *R* of a) Noah, b) Noah-MP and c) CLM4.

[Figure]

**Figure 6.** Boxplots of rainfall evaluation results of a) *R* and b) *RMSE*: minimum, maximum, 0.25, 0.50 and 0.75 percentiles, and outliers (red cross).

[Figure]

**Figure 7.** Threshold plots. For Noah (a, d, g, j), Noah-MP (b, e, h, k), and CLM4 (c, f, i, l) land surface schemes under three Slope angle Groups (S.G.) with S.G. 1 = 0.4-1.86º; S.G. 2 = 1.87-9.61º; S.G. 3 = 9.52-40.43º.

[Figure]

**Figure 8.** d-scores.

[Figure]

**Figure 9.** ROC curve for the calculated thresholds using different exceedance probability levels (for Noah-MP at the surface layer). The *no gain* line and the optimal performance point (the red point) are also presented.

[Figure]

**Figure 10.** The cross-validation of spatially distributed WRF soil moisture against the in-situ soil moisture observation at the single point soil moisture sensor in plain area: a) grid numbers shown on the slope map, b) correlation spatial performance.

[Figure]

**Figure 11.** The soil moisture comparisons of Grid 27 with the adjacent grids (16, 28, 26, 37).